# Microglial SIRPα regulates the emergence of CD11c+ microglia and demyelination damage in white matter

Miho Sato-Hashimoto[1], Tomomi Nozu[1], Riho Toriba[1], Ayano Horikoshi[1], Miho Akaike[1], Kyoko Kawamoto[1], Ayaka Hirose[1], Yuriko Hayashi[1], Hiromi Nagai[1], Wakana Shimizu[1], Ayaka Saiki[1], Tatsuya Ishikawa[2,3,4], Ruwaida Elhanbly[2,3,4,5], Takenori Kotani[6], Yoji Murata[6], Yasuyuki Saito[6], Masae Naruse[7], Koji Shibasaki[7], Per-Arne Oldenborg[8], Steffen Jung[9], Takashi Matozaki[6], Yugo Fukazawa[2,3,4], Hiroshi Ohnishi[1]*

[1]Department of Laboratory Sciences, Gunma University Graduate School of Health Sciences, Gunma, Japan; [2]Division of Brain Structure and Function, Faculty of Medical Sciences, University of Fukui, Fukui, Japan; [3]Research Center for Child Mental Development, Faculty of Medical Sciences, University of Fukui, Fukui, Japan; [4]Life Science Innovation Center, University of Fukui, Fukui, Japan; [5]Department of Anatomy, Histology and Embryology, Faculty of Veterinary Medicine, Assiut University, Asyut, Egypt; [6]Division of Molecular and Cellular Signaling, Department of Biochemistry and Molecular Biology, Kobe University Graduate School of Medicine, Kobe, Japan; [7]Department of Molecular and Cellular Neurobiology, Gunma University Graduate School of Medicine, Gunma, Japan; [8]Department of Integrative Medical Biology, Section for Histology and Cell Biology, Umeå University, Umeå, Sweden; [9]Department of Immunology, Weizmann Institute of Science, Rehovot, Israel

*For correspondence:
ohnishih@gunma-u.ac.jp

Competing interests: The authors declare that no competing interests exist.

**Abstract** A characteristic subset of microglia expressing CD11c appears in response to brain damage. However, the functional role of CD11c+ microglia, as well as the mechanism of its induction, are poorly understood. Here we report that the genetic ablation of signal regulatory protein α (SIRPα), a membrane protein, induced the emergence of CD11c+ microglia in the brain white matter. Mice lacking CD47, a physiological ligand of SIRPα, and microglia-specific SIRPα-knockout mice exhibited the same phenotype, suggesting that an interaction between microglial SIRPα and CD47 on neighbouring cells suppressed the emergence of CD11c+ microglia. A lack of SIRPα did not cause detectable damage to the white matter, but resulted in the increased expression of genes whose expression is characteristic of the repair phase after demyelination. In addition, cuprizone-induced demyelination was alleviated by the microglia-specific ablation of SIRPα. Thus, microglial SIRPα suppresses the induction of CD11c+ microglia that have the potential to accelerate the repair of damaged white matter.
DOI: https://doi.org/10.7554/eLife.42025.001

## Introduction

Microglia constantly survey the microenvironment of the brain. When microglia encounter tissue damage, they become activated, produce multiple humoral factors and show enhanced phagocytic activity; microglia thus play a central role in the removal and repair of damaged tissues (*Kettenmann et al., 2011*). However, the excessive activation of microglia results in the progression

of inflammation and tissue degeneration, which is harmful to the brain environment (*Kettenmann et al., 2011*). The activation of microglia has been reported in various neurodegenerative diseases and brain injuries, including Alzheimer's disease (AD), amyotrophic lateral sclerosis (ALS), and demyelinating diseases (*Chiu et al., 2013*; *Holtman et al., 2015*; *Kamphuis et al., 2016*; *Remington et al., 2007*; *Wang et al., 2015*).

In animal models of these disorders, a characteristic subset of microglia that express integrin αX or CD11c, a marker of murine peripheral dendritic cells and selected macrophages, appear in the affected brain regions. The expression of CD11c is a marker for 'primed' microglia that are not fully activated but rather are in a pre-activation state (*Holtman et al., 2015*; *Norden and Godbout, 2013*). The appearance of CD11c$^+$ microglia has also been reported during postnatal development and normal aging (*Bulloch et al., 2008*; *Kaunzner et al., 2012*). In both cases, CD11c$^+$ microglia characteristically appeared in the brain white matter, where myelin construction during the late developmental stage (*Baumann and Pham-Dinh, 2001*) or the accumulation of damaged myelin debris during aging were remarkable (*Safaiyan et al., 2016*). It was also shown that demyelination markedly induced CD11c$^+$ microglia, even in the adult brain (*Remington et al., 2007*), but this induction was suppressed in mutant mice lacking Trem2 or Cx3Cr1 (*Lampron et al., 2015*; *Poliani et al., 2015*), which are functional molecules that promote phagocytosis. Of note, the clearance of myelin debris was markedly impaired in these mutant mice (*Cantoni et al., 2015*; *Lampron et al., 2015*; *Poliani et al., 2015*). CD11c$^+$ microglia also accumulate around amyloid plaques in an AD mouse model that has gene expression patterns indicating an enhanced capacity for phagocytosis (*Kamphuis et al., 2016*; *Wang et al., 2015*). The phagocytic ability of CD11c$^+$ microglia thus might contribute to the removal and repair of damaged tissues. However, the physiological roles of CD11c$^+$ microglia, as well as the mechanisms controling the induction of CD11c$^+$ microglia, are not fully understood.

SIRPα (CD172a) is a membrane protein that is highly expressed in macrophages and dendritic cells. The extracellular region of SIRPα specifically interacts with another membrane protein CD47 to promote cell–cell contact signals (*Matozaki et al., 2009*). Interactions between SIRPα on phagocytic cells and CD47 on phagocytosed targets, such as erythrocytes, cancer cells, and apoptotic cells, act as a 'don't eat me' signal and thereby control erythrocyte homeostasis, elimination of cancer cells, and the formation of arteriosclerotic plaques (*Chao et al., 2011*; *Ishikawa-Sekigami et al., 2006*; *Kojima et al., 2016*; *Willingham et al., 2012*). In the brain, SIRPα and CD47 are abundantly expressed on neurons (*Ohnishi et al., 2010*), and SIRPα is also expressed on microglia (*Gitik et al., 2011*). It is assumed that SIRPα on microglia interacts with neuronal CD47 to suppress microglial activation (*Saijo and Glass, 2011*), because SIRPα contains an immunoreceptor tyrosine-based inhibitory motif (ITIM) in its cytoplasmic region (*Barclay and Hatherley, 2008*; *Matozaki et al., 2009*). However, direct in vivo evidence supporting this model is missing. Here, we analysed the role of SIRPα in the control of microglial activation using CD47–SIRPα signal-deficient mice and defined a critical microglia control module. Specifically, our results establish that SIRPα on microglia controls cell activation through the interaction with its ligand CD47 and negatively regulates the induction of CD11c$^+$ microglia, which have the potential to support the repair of damaged myelin, in white but not in grey matter.

## Results

### Emergence of CD11c$^+$ microglia in the brain white matter of SIRPα-deficient mice

To examine the role of SIRPα in microglial activation, brains of SIRPα knockout (KO) mice were subjected to immunohistochemical analysis using antibodies specific to Iba1, a microglia marker, and to CD68, a marker for phagocytically active microglia (*Figure 1A*). The numbers of Iba1$^+$ cells as well as of Iba1$^+$/CD68$^+$ cells in the white matter, such as the fimbria, were significantly increased in the brains of SIRPα KO mice compared with those of wild-type (WT) control mice, suggesting the activation of microglia in these regions (*Figure 1A and B*). The activation of microglia in the white matter of SIRPα KO mice was similar to the phenotype reported in aged mice, in which numbers of CD11c$^+$ microglia were reported to be increased (*Kaunzner et al., 2012*). Next, we examined the effect of the genetic ablation of SIRPα on the expression of CD11c by microglia by using CD11c–EYFP

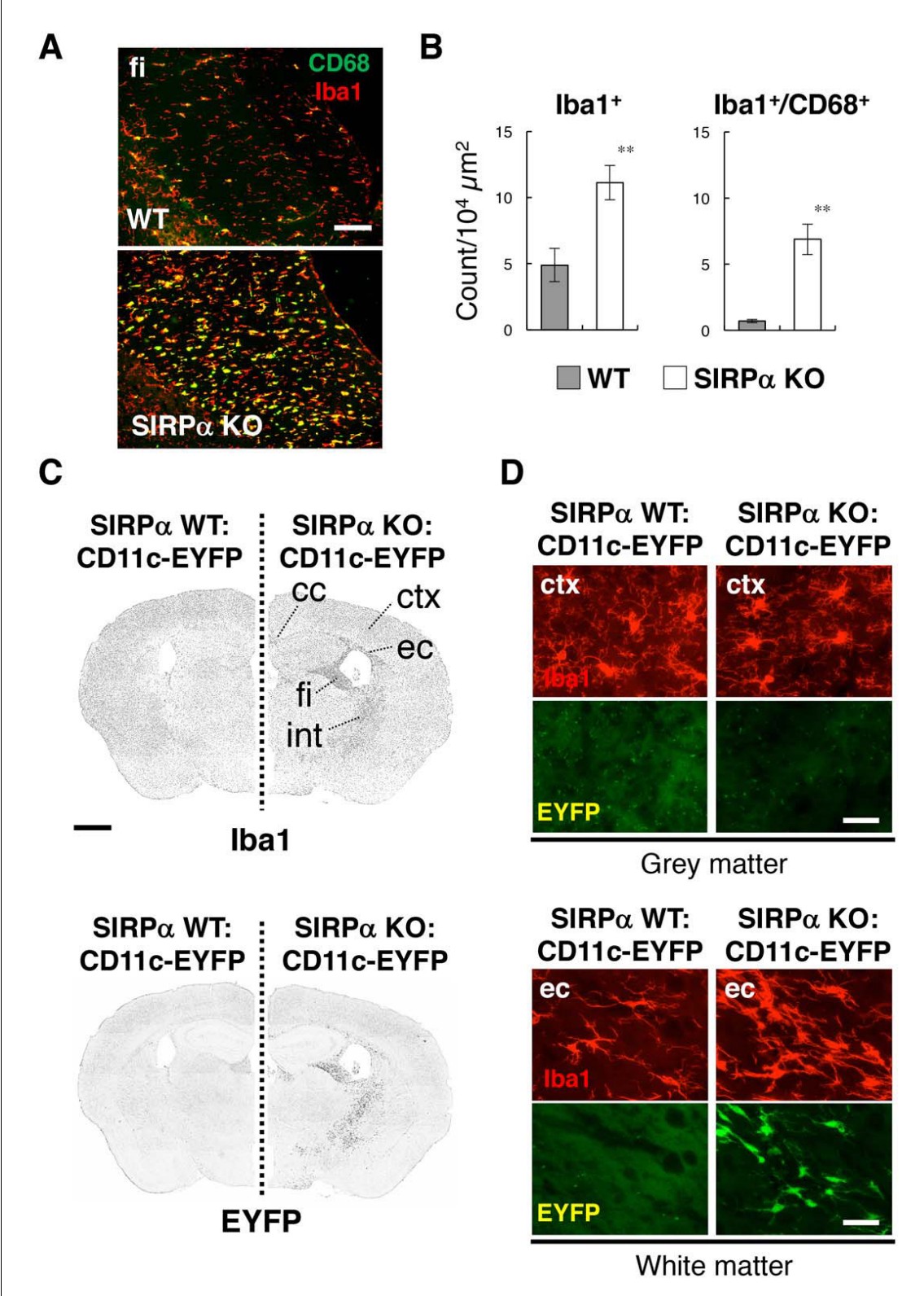

**Figure 1.** Activation of microglia in the brain white matter of SIRPα KO mice. (A) Immunofluorescence staining of coronal brain sections prepared from control (WT) or SIRPα null mutant mice (SIRPα KO) with antibodies to Iba1 (red) and CD68 (green). Merged images are shown. fi, fimbria. (B) Quantitative analysis of the number of Iba1$^+$ (*left panel*) and Iba1$^+$/CD68$^+$ (*right panel*) microglia in the fimbria of WT (filled bars) and SIRPα KO mice (open bars) at 18–20 wks of age. Data are the means ± SEM (n = 6 images from three mice for each genotype). **p<0.01 (Welch's t-test). (C, D)

*Figure 1 continued on next page*

*Figure 1 continued*

Immunofluorescence staining with a specific antibody for Iba1 of coronal brain sections prepared from control CD11c–EYFP Tg mice (at 30–32 wks of age) carrying a homozygous SIRPα WT allele (SIRPα WT:CD11c–EYFP, *left side*) or a SIRPα null mutation (SIRPα KO:CD11c–EYFP, *right side*). Images in panel (C) are lower-magnification images of the immunoreactivity of Iba1 (*upper panels*) and of the fluorescence of EYFP in the same sections (*lower panels*). Images were converted to grey scale and then inverted to clarify the fluorescence signal. Images in panel (D) are higher-magnification images of the immunoreactivity of Iba1 (*red*) and fluorescence of EYFP (*green*) in grey matter (ctx, cerebrum cortex; *upper panels*) and white matter (ec, external capsule; *lower panels*). cc, corpus callosum; ec, external capsule; fi, fimbria; int, internal capsule. Data are representative of at least three independent animals. Scale bars: 100 µm (A), 1 mm (C), and 50 µm (D).

DOI: https://doi.org/10.7554/eLife.42025.002

transgenic (Tg) mice (*Lindquist et al., 2004*), in which the expression of CD11c is detected with the enhanced yellow fluorescent protein reporter gene. The numbers of Iba1[+] cells and EYFP[+] cells were markedly increased in the white matter (including the corpus callosum, external capsule, fimbria, and internal capsule) of SIRPα KO:CD11c–EYFP Tg mice when compared with that of control SIRPα WT:CD11c-EYFP Tg mice (*Figure 1C*). Most EYFP[+] cells in the white matter of SIRPα KO mice were Iba1 positive, suggesting that they were CD11c[+] microglia (*Figure 1D lower panels*). By contrast, the EYFP signal was barely detected in the grey matter of either genotype (*Figure 1D upper panels*). The white matter-specific emergence of Iba1[+]/CD11c[+] cells was also confirmed using an antibody that is specific to CD11c in the brain or spinal cord of SIRPα KO mice (*Figure 2A and B*).

Myeline damage causes microglia activation and an increase in the number of CD11c[+] microglia in the white matter of the brain (*Remington et al., 2007*). Thus, we examined the structural integrity of myelin in SIRPαKO mice. Although a marked increase in the number of CD11c[+] cells was observed in the anterior commissure of SIRPαKO mice, no appreciable demyelination was evident after immunostaining for myelin basic protein (MBP) (*Figure 2C*). Myelination in the corpus callosum and fimbria were also normal in SIRPαKO mice when examined by myelin staining with a gold-phosphate complex (Black-Gold II) (*Figure 2D*).

## Increased expression of innate immune molecules in CD11c[+] microglia in the brain of SIRPα-deficient mice and aged mice

We then examined the characteristics of the CD11c[+] microglia. Microglia were isolated from the spinal cord of SIRPα KO and control WT mice, and the CD11b[+]/CD45[dim/lo] fraction was analysed by flow cytometry (*Figure 3A–3C*). The yield of microglia from SIRPα KO mice was significantly greater than that from control mice (means ± SEM were $8.02 \pm 0.36$ and $4.57 \pm 0.98 \times 10^4$ cells/mouse, respectively; n = 5 for control and 3 for KO animals; p=0.02161 (Welch's *t*-test)), indicating an increased number of microglia in the spinal cord of SIRPα KO mice. As expected, the expression of SIRPα was completely abolished in microglia prepared from SIRPα KO mice (*Figure 3A*). Consistent with the results from immunohistochemical analysis, CD11c[+]/CD11b[+]/CD45[dim/lo] microglia were increased in SIRPα KO mice when compared with control WT mice (*Figure 3B*). The forward (FSC) and side (SSC) scatter distribution of CD11c[+] microglia were similar to those of the CD11c[−] microglia in the SIRPα KO mice, suggesting that cell size and granularity (complexity) were comparable between the two subsets (*Figure 3—figure supplement 1*). CD11c[+] microglia expressed higher levels of innate immune molecules (including CD14, Dectin-1, and CD68) when compared with CD11c[−] microglia in SIRPα KO mice (*Figure 3C*). To compare the CD11c[+] microglia in SIRPα KO mice with those in aged WT mice (*Bulloch et al., 2008*; *Kaunzner et al., 2012*), microglia isolated from adult (16–18 weeks of age) and aged (69–105 weeks of age) WT mice were analysed in the same manner (*Figure 3D–F*). The expression levels of SIRPα in microglia were comparable in aged and adult mice (*Figure 3D*). CD11c[+] microglia were increased in aged mice (as in SIRPα KO mice) when compared with adult mice (*Figure 3E*). In addition, the CD11c[+] microglia in the aged mice expressed higher levels of CD14, Dectin-1, and CD68 than the CD11c[−] microglia of these mice, as was also the case in SIRPα KO mice (*Figure 3F*).

To address the effect of the increase of CD11c[+] microglia on inflammatory responses, we examined the expression of pro- and anti-inflammatory cytokines in the brain and spinal cord by quantitative PCR analysis. Among the examined cytokines (*Tnfa*, *Il1b*, *Il6*, *Il10*, and *Tgfb*), the expression of TNF-α (*Tnfa*) was increased in SIRPα KO as compared to WT mice (*Figure 3—figure supplement 2*).

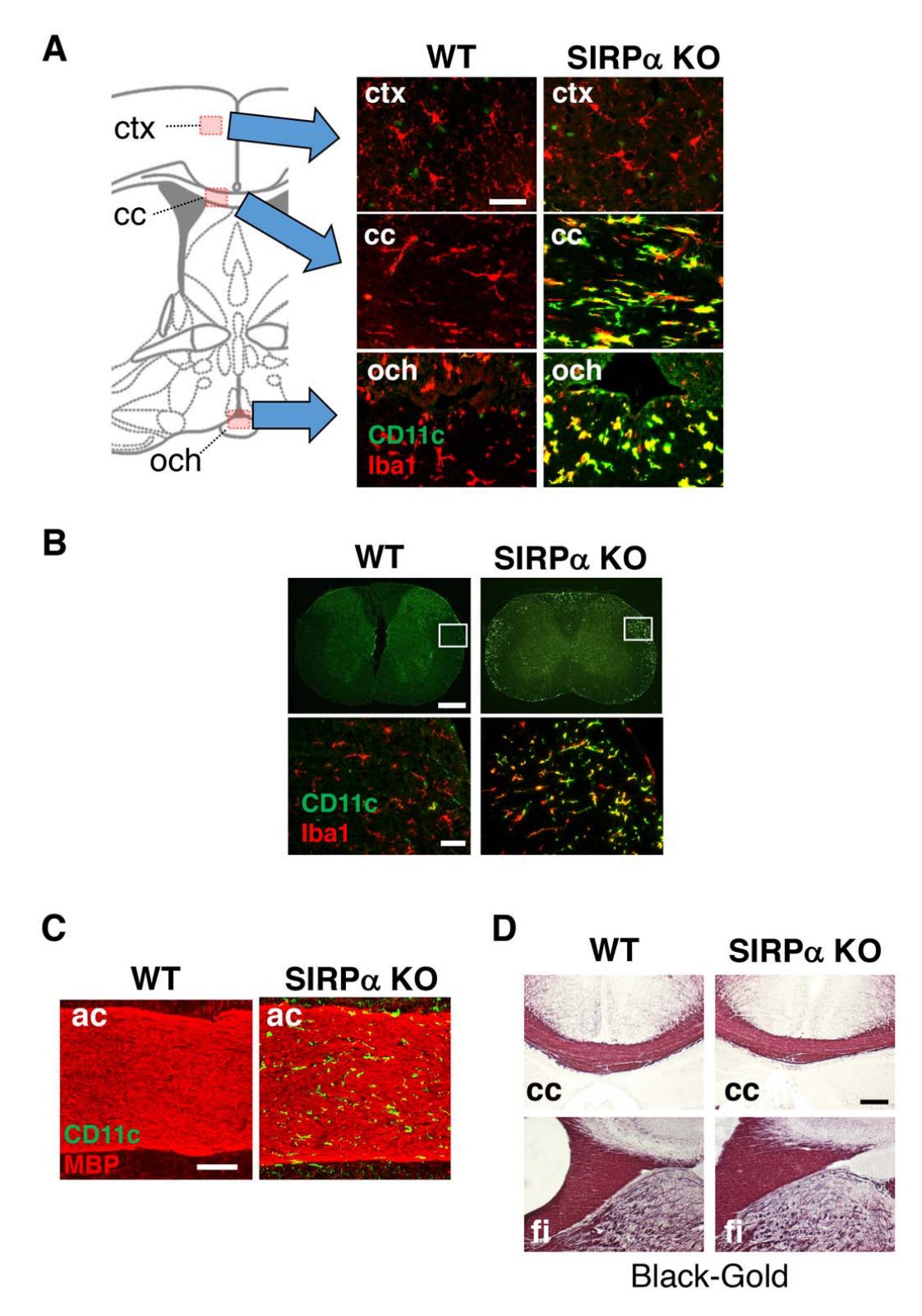

**Figure 2.** CD11c[+] microglia in the white matter of SIRPα KO mice. (A, B) Coronal sections were prepared from brain (A) and spinal cord (B) of control (WT) or SIRPα null-mutant mice (SIRPα KO) at 18–20 wks of age and stained with specific antibodies for Iba1 (*red*) and CD11c (*green*). In panel (A), merged images are shown. Schematic diagrams of brain sections are shown on the left side. The boxed areas in panel (B) are shown at a higher magnification in the lower panels. cc, corpus callosum; ctx, cerebrum cortex; och, optic chiasma. (C, D) Coronal brain sections of control (WT) or SIRPα

*Figure 2 continued*

null-mutant mice (SIRPα KO) at 29–32 (C) or 10–11 (D) wks of age were stained with antibodies to myelin basic protein (MBP) (*red*) and CD11c (*green*) (C), or Black-Gold (D). ac, anterior commissure; cc, corpus callosum; fi, fimbria. Data are representative of at least three independent animals. Scale bars: 50 μm (A, B *lower panels*), 1 mm (B *upper panels*), 100 μm (C), and 200 μm (D).
DOI: https://doi.org/10.7554/eLife.42025.003

## Induction of CD11c⁺ microglia in the white matter of CD47-deficient mice

To address the mechanism involved in the regulation of microglia activation by SIRPα, we examined the effect of the genetic ablation of CD47, a membrane protein and SIRPα ligand (*Matozaki et al., 2009*). In the brain of CD47 KO mice, CD11c⁺ microglia were increased in the white matter, as was also the case for SIRPα KO mice (*Figure 4A*). The number of Iba1⁺ microglia was increased more than 2.5-fold in the hippocampal fimbria, and about 70% of the Iba1⁺ microglia expressed CD11c (*Figure 4B*). As in SIRPα KO mice, the yield of microglia in CD47 KO mice was increased compared to that in WT mice (means ± SEM were $7.65 ± 0.92$ and $3.34 ± 0.85 × 10^4$ cells/head, respectively; n = 5 animals; p=0.00874 (Welch's t-test)). Flow cytometric analysis showed that CD47 was expressed in WT microglia and was completely absent in CD47 KO microglia (*Figure 4C*). In addition, we found that the expression of SIRPα was markedly increased in CD47 KO microglia (*Figure 4C*). We also examined the expression of CD47 in SIRPα KO microglia, and found that the expression of CD47 was increased in microglia prepared from SIRPα KO mice (*Figure 4D*).

## Increased expression of genes for the repair of damaged myelin in CD47-deficient mice

To address the impact of the emergence of activated CD11c⁺ microglia on the brain environment, we compared gene expression in the white matter (optic nerve and optic tract) of CD47 KO mice and WT control mice by microarray transcriptome analysis. The expression of total of 14,875 genes was detected in both or either of the genotypes. The expression of 594 and 548 genes was markedly (>2 fold) increased and decreased, respectively, in CD47 KO mice when compared with WT mice (*Supplementary file 1*). Pathway analysis with Database for Annotation, Visualization and Integrated Discovery (DAVID) ver. 6.8 (https://david.ncifcrf.gov/) (*Huang et al., 2009a*; *Huang et al., 2009b*) suggested that the genes whose expression increased in the white matter of CD47 KO mice were significantly enriched in pathways such as infectious diseases, phagocytosis, and immune responses, probably reflecting the activation of microglia (*Figure 5A*). By contrast, genes whose expression was decreased in CD47 KO mice were mostly enriched in pathways for neuronal ligand–receptor interaction, and were also enriched in pathways such as calcium signaling and neuronal synapses (*Figure 5B*).

Consistent with the results of immunohistochemical and flow cytometric analyses, the expression of *Itgax* (CD11c), *Clec7a* (Dectin-1), *Cd68*, and *Cd14* were markedly increased (>2-fold: Log2 ratio >1) in the white matter of CD47 KO mice compared with that of WT mice (*Figure 5C*). By contrast, the expression of the microglial markers *Aif1* (Iba1) and *Itgam* (CD11b) was only moderately increased (<2 fold). Thus, the marked increase in the expression of CD11c and the other molecules was probably related to characteristic changes in the mutant microglia (or other cells) in the white matter, rather than to the increased number of microglia. Quantitative PCR analysis of selected genes (*Itgax, Igf1, Trem2, Ccl3*), which were increased in the microarray analysis, also showed significant induction (*Figure 5—figure supplement 1*). We noted that several transcripts whose expression was reported to be increased in microglia during the repair of damaged myelin (*Holtman et al., 2015*; *Olah et al., 2012*; *Poliani et al., 2015*; *Selvaraju et al., 2004*) were markedly increased (>2 fold) in the white matter of CD47 KO mice (*Figure 5C*). These included *Itgax, Lgals3*, and *Clec7a*, markers of primed microglia; *Lpl, Apoc1*, and *Ch25h*, encoding molecules for lipid transport; *Mmp12*, encoding a tissue remodeling factor; *Igf1*, and *Spp1*, encoding trophic factors promoting oligodendrocyte differentiation; and *Cst7* (Cystatin F), which encodes a cysteine proteinase inhibitor. We also found that several positive regulators of microglia phagocytosis, *Trem2, Tyrobp*, and *Cx3cr1* (*Lampron et al., 2015*; *Poliani et al., 2015*), were increased in the white matter of the mutant mice.

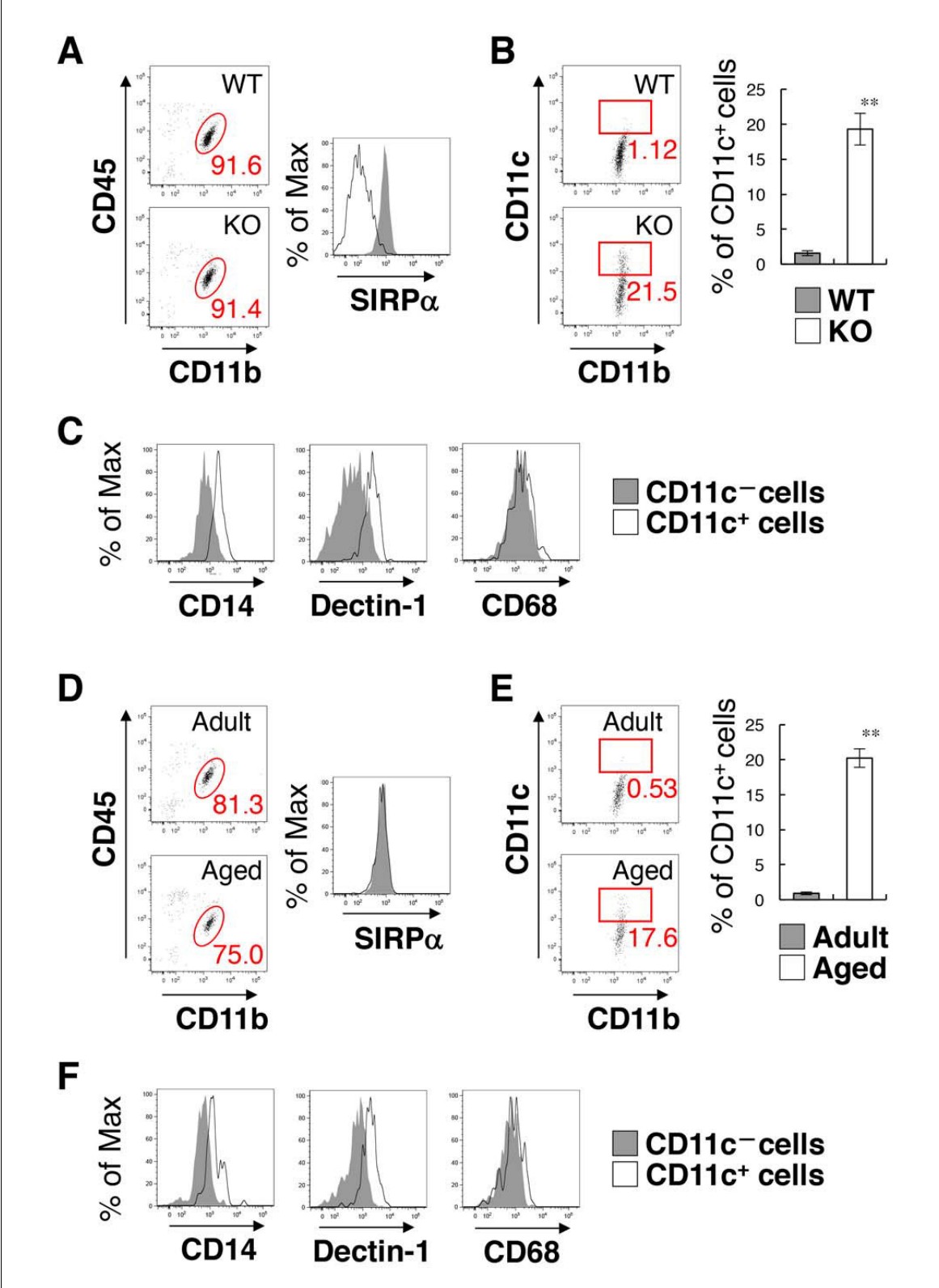

**Figure 3.** Flow cytometry analysis of microglia in SIRPα KO mice and WT aged mice. (**A**) Cells isolated from the spinal cord of control WT or SIRPα KO (KO) mice at 14 wks of age were incubated with a PE-conjugated monoclonal antibody (mAb) to SIRPα, a PerCP–Cy5.5-conjugated mAb to CD45, and an FITC-conjugated mAb to CD11b. The expression of CD11b and CD45 on monocyte cells or of SIRPα on CD11b[+]/CD45[dim/lo] microglia was analysed by flow cytometry. The percentage of CD11b[+]/CD45[dim/lo] microglia among putative monocytes is indicated in each plot (*left plots*). The expression

*Figure 3 continued on next page*

*Figure 3 continued*

profiles for SIRPα on CD11b⁺/CD45^dim/lo microglia are shown in the right panel. (B) Cells prepared as in panel (A) from WT or SIRPα KO mice at 14–21 wks of age were stained with antibodies to CD45, CD11b, and CD11c, and analysed by flow cytometry. The percentage of CD11b⁺/CD45^dim/lo/CD11c⁺ microglia among total CD11b⁺/CD45^dim/lo microglia is indicated in each plot. Quantitative data are shown in the right panel. Filled and open bars indicate WT and SIRPα KO mice, respectively. (C) Cells prepared from SIRPα KO mice were stained for CD45, CD11b and CD11c, as well as for CD14, Dectin-1, or CD68. Plots were gated on CD11b⁺/CD45^dim/lo cells, and CD14, Dectin-1, or CD68 on CD11c–positive and -negative microglia were analysed. The expression profiles for each molecule in CD11b⁺/CD45^dim/lo microglia are shown. (D) Cells isolated from adult (18 wks of age) or aged (104 wks of age) mice were isolated and analysed as in panel (A). (E) Cells prepared as in panel (D) from adult (16–18 wks of age) or aged (69–105 wks of age) mice were analysed as in (B). Quantitative data are shown in the right panel. Filled and open bars indicate adult and aged mice, respectively. (F) Cells prepared from aged mice were analysed as in panel (C).Data in panels (B) and (E) are the means ± SEM ($n$ = 5 (B) and 3 (E) independent experiments). **$p<0.01$ (Welch's t-test). Other data are representative of at least 3–5 independent experiments. Filled and open traces in panels (A), (C, F), and (D) indicate WT and SIRPα KO mice (A), CD11c⁻ and CD11c⁺ cells (C, F), and adult and aged mice (D), respectively.
DOI: https://doi.org/10.7554/eLife.42025.004

The following figure supplements are available for figure 3:

**Figure supplement 1.** Forward (FSC) and side (SSC) scatter distribution of CD11c⁺microglia.
DOI: https://doi.org/10.7554/eLife.42025.005

**Figure supplement 2.** Expression of pro- and anti-inflammatory cytokines in the brain and spinal cord of SIRPα KO mice.
DOI: https://doi.org/10.7554/eLife.42025.006

We next examined gene expression in the brain mononuclear cells, in which microglia were enriched. Expression of a total 16,544 genes was detected in either CD47 KO and WT control cells or both. Among them, the expression of 1323 and 2286 genes was markedly (>2 fold) increased and decreased, respectively, in the CD47 KO cells (*Supplementary file 2*). Genes whose expression increased in CD47 KO brain cells were significantly enriched in pathways for T cell receptor signaling, axon guidance, proteoglycans in cancer, TNF signaling, and NF-κB signaling (*Figure 5A*); genes whose expression decreased in CD47 KO brain cells were enriched in cancer-associated pathways including Wnt, Hippo, and Rap1 signaling, as well as cardiomyopathy (*Figure 5B*).

Comparison of array data revealed 32 and 55 genes that were commonly increased and decreased (>2 fold), respectively, in both the white matter and the brain mononuclear cells of CD47 KO mice (*Figure 5—figure supplement 2*). Shared induced genes included myelin-repair related genes, such as *Itgax*, *Igf1*, *Lpl*, *Apoc1*, *Ch25h*, *Mmp12*, *Spp1*, and *Cst7* (*Figure 5C*). The expression of *Clec7a*, *Cd68*, *Trem2*, and *Cx3cr1*, which were markedly increased in the white matter of CD47 KO mice, showed an only moderate (<2 fold) increase in the brain mononuclear cells (*Figure 5C*). Substantially higher levels of expression of these genes in microglia might mask the increased expression of these genes in the limited CD11c⁺ subset, which was only ~5% of the total microglia prepared from the whole brain of CD47 KO mice (data not shown).

## Induction of CD11c⁺ microglia in the brain white matter of microglia-specific SIRPα-deficient mice

To examine the cell type involved in the suppression of CD11c⁺ microglia by SIRPα, we analysed microglia-specific SIRPα conditional KO (cKO) mice that were generated by crossing SIRPα-flox (Sirpa^flox/flox) mice (*Washio et al., 2015*) with Cx3cr1-CreER^T2 (*Cx3cr1^tm2.1(cre/ERT2)Jung*) mice to achieve microglia-specific gene targeting (*Goldmann et al., 2013*; *Safaiyan et al., 2016*; *Wolf et al., 2013*). The Cx3cr1–CreER^T2 transgene induces tamoxifen (TAM)-dependent rearrangement of a floxed gene not only in microglia but also in peripheral myeloid cells (monocytes and dendritic cells) and tissue macrophages (Kupffer cells). However, the peripheral cells are short-lived and are replaced by their progeny without rearranged gene within a few weeks after administraiton of TAM (*Goldmann et al., 2013*). Indeed, SIRPα was markedly reduced in the white pulp of SIRPα cKO mouse spleen after TAM-treatment, but recovered to the normal level after several weeks (*Supplementary file 3*). This recovery is probaly due to the replacement of certain types of conventional dendritic cells (cDCs) or macrophages, in which the SIRPα-floxed gene was rearranged, with their progeny in which the gene is not rearranged. By contrast, the rearranged gene remained stable in microglia (*Goldmann et al., 2013*) because of their long lifespan of around 15 months (*Füger et al., 2017*). As a result, microglia-specific gene targeting is achieved in Cx3cr1–CreER^T2

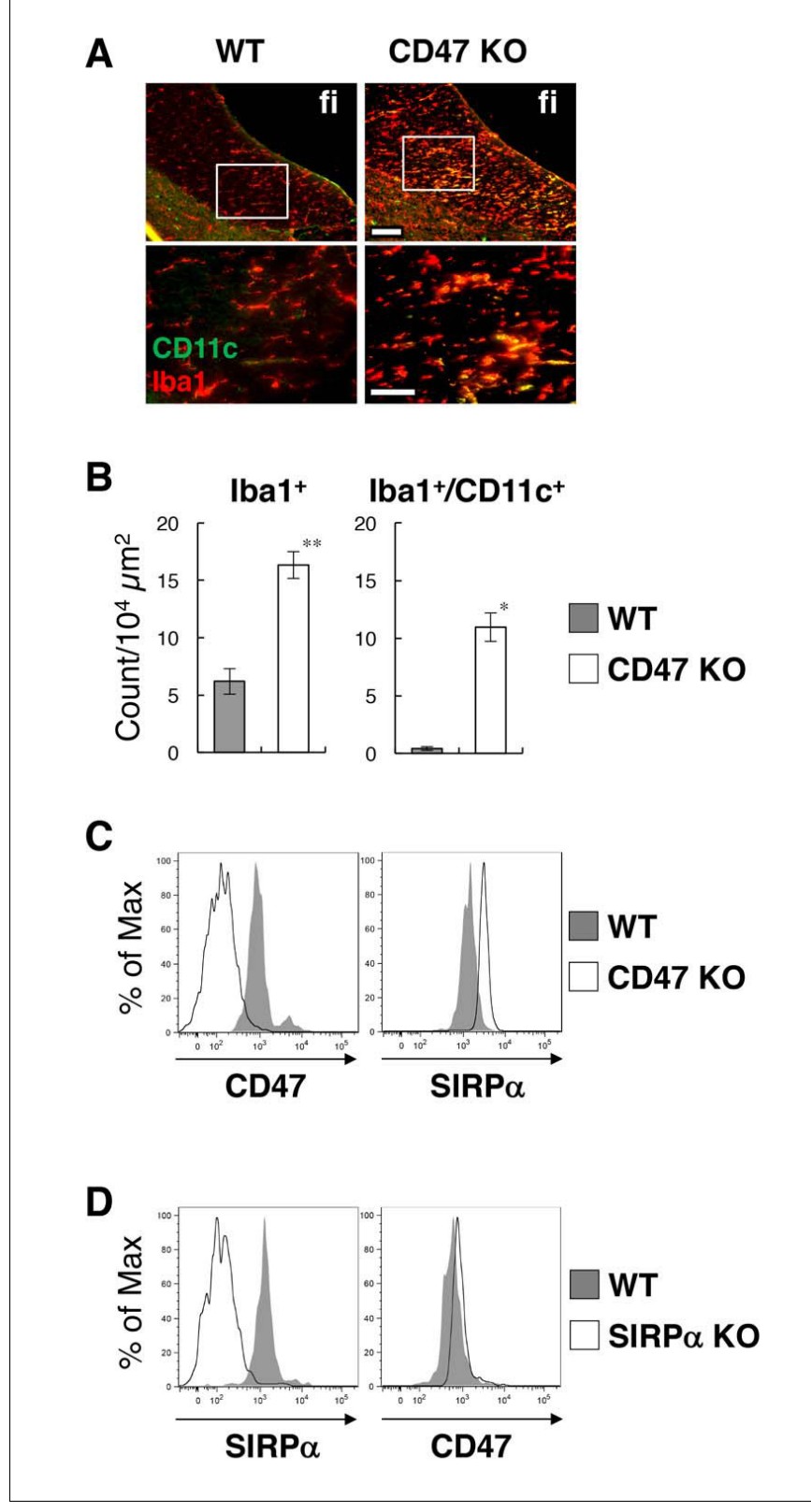

**Figure 4.** Activation of microglia in the brain white matter of CD47 KO mice. (**A**) Immunofluorescence staining of coronal brain sections prepared from control (WT) or CD47 KO mice at 19 wks of age with antibodies to Iba1 (*red*) and CD11c (*green*). Merged images are shown. The boxed areas in the upper panels are shown at higher magnification in the lower panels. fi, fimbria. Scale bars: 100 μm (*upper panels*), 50 μm (*lower panels*). (**B**)
*Figure 4 continued on next page*

*Figure 4 continued*

Quantitative analysis of the number of Iba1[+] (*left panel*) and Iba1[+]/CD11c[+] (*right panel*) microglia in the fimbria of WT (filled bars) and CD47 KO mice (open bars) at 13–27 wks of age. Data are the means ± SEM (n = 3 images from 3 mice for each genotype). **p<0.01, *p<0.05 (Welch's t-test). (C, D) Cells were isolated from the spinal cord of WT or CD47 KO mice at 14–16 wks of age (C), or from brain of WT or SIRPα KO female mice at 12 wks of age (D). Expressions of SIRPα and CD47 on CD11b[+]/CD45[dim/lo] microglia were analysed by flow cytometry. Filled and open traces indicate WT and CD47 KO (C) or SIRPα KO (D) mice, respectively. Data in (C) are representative of at least three independent experiments. Data in panel (D) are also representative of th independent experiments with brain (n = 2, one male pair and one female pair) and spinal cord (n = 1, one female pair) microglia.
DOI: https://doi.org/10.7554/eLife.42025.007

mice several weeks after TAM treatment (*Goldmann et al., 2013*). Thus, SIRPα cKO mice were analyzed more than 8 weeks after TAM treatment in our experiments.

Flow-cytometric analysis revealed that more than 98% of microglia were SIRPα-negative in the brain of tamoxifen-treated SIRPα-flox:Cx3cr1–CreER[T2] mice (*Figure 6A*), and that the numbers of Iba1[+], Iba1[+]/CD11c[+], and Iba1[+]/CD68[+] microglia were increased in the hippocampal fimbria of these cKO mice (*Figure 6B and C*). Iba1[+]/CD11c[+] cells were also detected in other white-matter regions of the brain and spinal cord in SIRPα cKO, but not in control SIRPα-flox:— mice, and were barely detected in the grey matter of both genotypes (*Figure 6D and E*). Flow cytometric analysis in SIRPα cKO mice showed that, when compared with CD11c[−] microglia, CD11c[+] microglia expressed higher levels of CD14, Dectin-1, and CD68 (*Figure 6—figure supplement 1*). All of these results were very similar to those observed in SIRPα null KO mice, and suggest that a lack of the interaction between SIRPα on microglia and CD47 on neighbouring cells is the primary cause of the induction of CD11c[+] microglia in CD47–SIRPα signal-deficient mice.

A small subset of resident CD11c[+] microglia has been reported in normal mouse brain (*Bulloch et al., 2008*). Thus, the marked increase in the number of CD11c[+] microglia in CD47–SIRPα-deficient mice could be due to the expansion of the small subset of resident CD11c[+] microglia. To address this, we examined CD11c[+] cell-specific SIRPα cKO mice that were generated by crossing SIRPα-flox mice and CD11c-Cre mice (*Washio et al., 2015*). In these mice, Cre is expressed under the control of the CD11c promoter, and recombination of the SIRPα-floxed gene is induced in CD11c-expressing cells, such as dendritic cells and monocytes in the immune system. By contrast, CD11c-Cre mice exhibit very low recombination in the majority of microglia that are CD11c negative (*Goldmann et al., 2013*), and thus, in the brain, genetic ablation of SIRPα was achieved in the small subset of resident CD11c[+] microglia but not in the majority of microglia. However, the emergence of CD11c[+] microglia was not observed in the CD11c[+] cell-specific SIRPα cKO mice (*Figure 6—figure supplement 2*), suggesting that the expansion of resident CD11c[+] microglia was not induced by the lack of SIRPα. It is likely that the ablation of SIRPα in CD11c-negative microglia induced the expression of CD11c in these cells.

## Alleviation of cuprizone-induced demyelination in the brain white matter of microglia-specific SIRPα-deficient mice

Although myelin damage induces CD11c[+] microglia (*Remington et al., 2007*), demyelination was not observed by light microscopy in SIRPα null KO mice (*Figure 2C and D*). We examined the myelin structure again in microglia-specific cKO mice using electron microscopy. In the cross-section of the anterior commissure (*Figure 7A*), the frequency of myelinated axons and the g-ratio (the ratio of the inneraxonal diameter to the total outer diameter) (*Figure 7B*) were comparable between SIRPα cKO and control mice (frequency of myelinated axons 34.6 ± 4.37% for control (n = 3), 36.6 ± 4.05% for cKO (n = 3), p=0.759, Welch's t-test; g-ratio 0.63 ± 0.026 for control (n = 3), 0.65 ± 0.009 for cKO (n = 3), p=0.448, Welch's t-test). A scatter plot of all g-ratio data as a function of axon diameters was also comparable between the two genotypes (*Figure 7C*). These data suggest that the myelin structure was normal in the mutant mice.

We further examined the progression of white matter damage in microglia-specific SIRPα cKO mice undergoing cuprizone (Cpz)-induced demyelination (*Gudi et al., 2014*; *Norkute et al., 2009*). Feeding with Cpz, a copper chelator, for 5 weeks induced demyelination accompanied by the robust accumulation of microglia (microgliosis) in white matter regions (including the hippocampal alveus

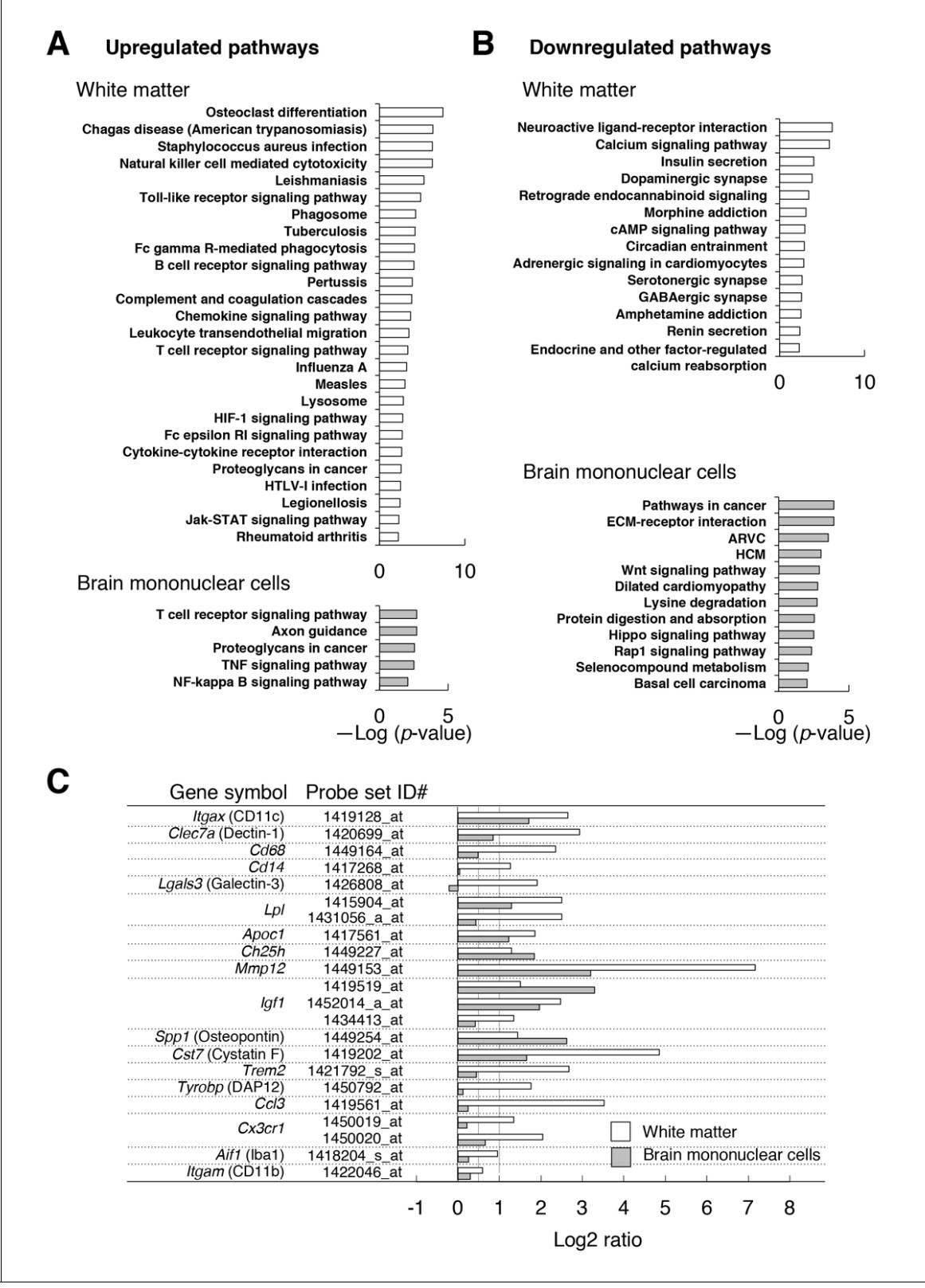

**Figure 5.** Microarray transcriptome analyses of the white matter and the brain mononuclear cells of CD47 KO mice. (**A,B**) The results of Kyoto Encyclopedia of Genes and Genomes (KEGG) pathway analysis with DAVID. Statistically significant (p-value <0.01) KEGG enrichment pathways of up- (**A**) or downregulated (**B**) genes in the white matter (optic nerve and optic tract) (*upper panels*) or in the brain mononuclear cells (*lower panels*) of CD47 KO mice. Enrichment score is expressed as –Log (*p*-value). ARVC, Arrhythmogenic right ventricular cardiomyopathy; HCM, Hypertrophic

*Figure 5 continued on next page*

*Figure 5 continued*

cardiomyopathy. (C) The expression changes of selected genes that are characteristic of microglia or of de- and re-myelination processes. For each probe set on the microarray, the fold-change of gene expression in the CD47 KO mice compared with WT mice was expressed as a $\log_2$ ratio. Open and filled bars represent data for white matter and the brain mononuclear cells, respectively. Probe set ID#: Affymetrix probe set ID number for Mouse 430 2.0 Gnome Arrays.

DOI: https://doi.org/10.7554/eLife.42025.008

The following source data and figure supplements are available for figure 5:

**Figure supplement 1.** Quantitative PCR analyses of white matter RNA.

DOI: https://doi.org/10.7554/eLife.42025.009

**Figure supplement 2.** Genes whose expression levels are commonly changed from WT levels in both the white matter and the brain mononuclear cells of CD47 KO mice.

DOI: https://doi.org/10.7554/eLife.42025.010

**Figure supplement 2—source data 1.** Source data of *Figure 5—figure supplement 2*.

DOI: https://doi.org/10.7554/eLife.42025.011

and corpus callosum) in control mice as previously reported (*Figure 8A*) (*Gudi et al., 2014*; *Norkute et al., 2009*). At the site of microgliosis, CD11c$^+$/Iba1$^+$ microglia were markedly increased in number, even in control mice (*Figure 8A*). In SIRPα cKO mice, both demyelination and microgliosis were significantly reduced after the same treatment (*Figure 8A-C*). At this time point, SIRPα cKO mice also showed a tendency to reduce the size of the CD11c-positive area compared to that in control mice, although this change was not statistically significant (*Figure 8D*). At an early stage of Cpz treatment (after 3 wks feeding with Cpz), demyelination was not observed in control or SIRPα cKO mice (*Figure 8A*), but abnormally strong immunoreactivity of MBP was observed in these mice (*Figure 8A*). This was probably related to myelin damage, because the epitope of the anti-MBP antibody that we used (DENPVV) is similar to that of a myelin damage-detectable antibody (QDENPVV) (*Matsuo et al., 1998*). Consistently, significant microgliosis, as well as an increase in CD11c-positive area size, was observed in both genotypes after 3 wks feeding with Cpz (*Figures 8A, C and D*). At this stage, both of these parameters (sizes of Iba1-positive area (microgliosis) and CD11c-positive area) were comparable between the two genotypes (*Figures 8A, C and D*). The repair of demyelination was observed after feeding with normal chow for 2 wks (*Figure 8A*). At this time point, Iba1-positive (microgliosis) and CD11c-positive area sizes in white matter were still larger than those in the basal condition (without feeding of Cpz) in both genotypes (*Figures 8A, C and D*). No statistical difference between SIRPα cKO mice and control mice was found for these parameters, whereas the size of the Iba1$^+$ area showed the tendency to be reduced in SIRPα cKO when compared to that in control mice (*Figure 8C*).

In the white matter, the number of cells expressing Olig2$^+$, a pan-oligodendrocyte marker (*Nishiyama et al., 2009*), was significantly decreased after 3 wks of Cpz feeding, suggesting that demyelination was preceded by the loss of oligodendrocytes (*Figure 8A and E*). The number of Olig2$^+$ cells was increased after 5 wks of Cpz treatment and recovered to normal levels after another two weeks of feeding with normal chow. Throughout the de- and re-myelination process, no significant differences in the number of Olig2$^+$ cells were noted between control and SIRPα cKO mice.

## Discussion

Cell–cell interactions between SIRPα on microglia and CD47 on neurons have been proposed to suppress the activation of microglia (*Ransohoff and Cardona, 2010*). However, direct evidence for the function of this module in the physiological context had not yet been provided. In this study, we demonstrated that SIRPα is a key molecule for the suppression of microglia activation in vivo. Microglia-specific SIRPα cKO mice exhibited the same phenotype as SIRPα null KO mice. Therefore, SIRPα proteins that are expressed on microglia directly control microglia activation and the induction of CD11c$^+$ microglia in vivo. The emergence of CD11c$^+$ microglia in the brains of microglia-specific SIRPα cKO mice establishes that CD11c$^+$ microglia are derived from resident microglia in the brain, and are not recruited monocytes that are spared in the TAM-induced system (*Goldmann et al., 2013*; *Safaiyan et al., 2016*; *Wolf et al., 2013*). The characteristics of CD11c$^+$ microglia in our mutant mice were similar to those of the 'primed microglia' that have been observed in aging and

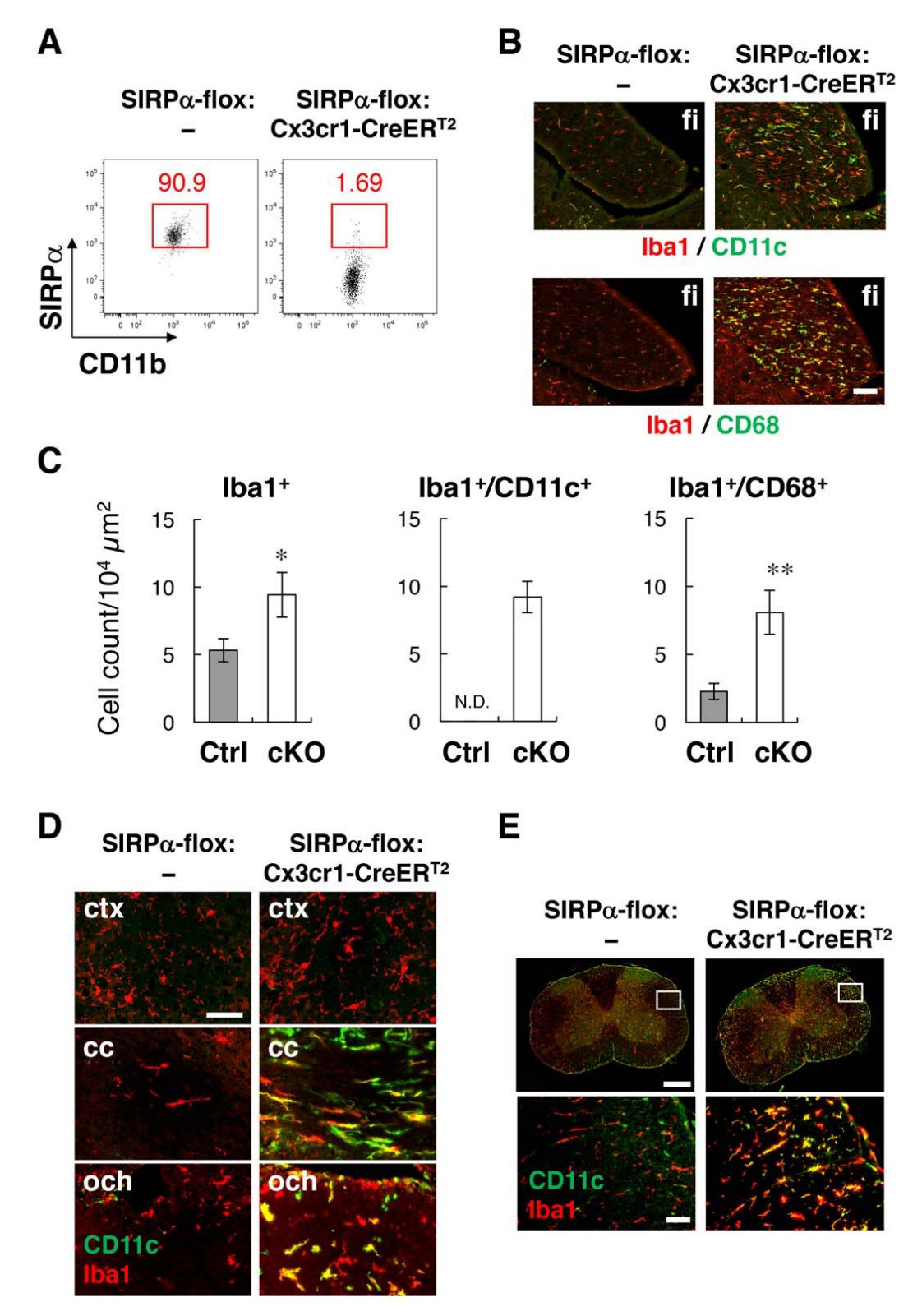

**Figure 6.** Phenotypes of microglia-specific SIRPα cKO mice. (**A**) Cells isolated from the brains of control (SIRPα-flox:—) or microglia-specific SIRPα cKO (SIRPα-flox:Cx3cr1–CreER[T2]) female mice at 11 wks of age were stained as in *Figure 3A*. The percentage of CD11b[+]/CD45[dim/lo]/SIRPα[+] or /SIRPα[−] microglia among total CD11b[+]/CD45[dim/lo] microglia is indicated in each plot. Data are representative of four independent experiments with microglia prepared from brain (n = 2) and spinal cord (n = 2). (**B**) Immunofluorescence staining of coronal brain sections prepared from control and SIRPα cKO
*Figure 6 continued on next page*

*Figure 6 continued*

mice at 22–30 wks of age with antibodies to Iba1 (*red*) and CD11c (*green in upper panels*) or CD68 (*green in lower panels*). Merged images are shown. (C) Quantitative analysis of the number of Iba1$^+$ (*left panel*) and Iba1$^+$/CD11c$^+$ (*middle panel*) and Iba1$^+$/CD68$^+$ (*right panel*) microglia in the fimbria of control (Ctrl) and SIRPα cKO (cKO) mice at 22–30 wks of age. Filled and open bars indicate control and SIRPα cKO mice, respectively. Data are the means ± SEM (n = 4–13 images from 2–6 mice for each genotype). *p<0.05, **p<0.01 (Welch's t-test). N.D., not detected. (D, E) Coronal sections were prepared from the brain (D) and spinal cord (E) of control or SIRPα cKO mice at 26–29 wks of age and stained with specific antibodies for Iba1 (red) and CD11c (*green*). Merged images are shown. The boxed areas in upper half of panel (E) are shown at a higher magnification in the lower half. cc, corpus callosum; ctx, cerebrum cortex; fi, fimbria; och, optic chiasma. Data in panels (B), (D), and (E) are representative of at least four independent animals. Scale bars: 100 μm (B), 50 μm (D, E *bottom images*), 1 mm (E *top images*).

DOI: https://doi.org/10.7554/eLife.42025.012

The following figure supplements are available for figure 6:

**Figure supplement 1.** Flow-cytometry analysis of microglia in SIRPα cKO mice.
DOI: https://doi.org/10.7554/eLife.42025.013

**Figure supplement 2.** Immunohistochemical analysis of CD11c$^+$ cell-specific SIRPα cKO mice.
DOI: https://doi.org/10.7554/eLife.42025.014

neurodegenerative diseases (*Holtman et al., 2015*); thus, SIRPα is a possible key component of the regulation of microglia priming in vivo. Molecules other than CD47, such as surfactant protein-A and -D (SP-A, SP-D), have also been reported as SIRPα ligands (*Matozaki et al., 2009*). However, we found that CD47 KO mice exhibited a phenotype similar to that of SIRPα KO mice, suggesting that the CD47–SIRPα interaction is indeed important for the suppression of microglia activation in the brain. Interestingly, the number of Iba1-positive cells in the fimbria of CD47 KO mice was greater than that in SIRPα null KO or cKO mice, whereas the number of CD11c/Iba1 double-positive cells was comparable in CD47 KO and SIRPα cKO mice (*Figure 1B*, *Figure 4B*, and *Figure 6C*). The CD47 signal may have a SIRPα-independent function that suppresses the proliferation of microglia.

We found that the cell-surface expression of SIRPα was significantly increased in microglia prepared from CD47 KO mice. Conversely, the expression of CD47 in microglia was also increased in SIRPα KO mice. Lack of interaction between CD47 and SIRPα may result in the upregulated expression of SIRPα or CD47 in microglia as a compensatory response, or in the stabilization of the CD47–SIRPα complex on microglia, because our previous study suggested that the interaction between CD47 and SIRPα induced endocytosis of the CD47–SIRPα complex in CHO cells (*Kusakari et al., 2008*).

It remains unclear which cell type expressed the CD47 that contributed to the suppression of CD11c$^+$ microglia. It has been proposed that CD47 that is expressed on neurons suppresses the activation of microglia through interactions with SIRPα (*Ransohoff and Cardona, 2010*). An in vitro study suggested that CD47 on myelin sheets interacted with SIRPα and suppressed microglial phagocytosis (*Gitik et al., 2011*). Another study reported that direct cell–cell contact between astrocytes and microglia suppressed the expression of CD11c on microglia (*Acevedo et al., 2013*). Thus, CD47 on oligodendrocytes or astrocytes may also participate in the suppression of microglia activation. In addition, microglia–microglia interaction through the CD47–SIRPα signal or cis-interaction between CD47 and SIRPα on the same microglia is another possibility. Supporting this, our preliminary results showed that microglial CD47 was increased in microglia-specific SIRPα cKO mice (*Supplementary file 4*). Further research with cell type-specific CD47 conditional KO mice is required to clarify which cell type is involved in the CD47-mediated effects.

CD11c$^+$ microglia are increased during neurodegenerative damage as well as in aging and during the postnatal development of normal brain (*Bulloch et al., 2008*; *Kaunzner et al., 2012*). We confirmed that CD11c$^+$ microglia in SIRPα KO mice had elevated expression of innate immune molecules, including CD14, Dectin-1, and CD68, which is characteristic of CD11c$^+$ microglia in WT aged mice. This suggested that the phenotype in the mutant mice was similar to the normal biological responses in the aged brain. Although the maturation-dependent downregulation of CD47 was shown in hematopoietic stem cells (*Jaiswal et al., 2009*), the age-dependent reduction of CD47 has not been reported in the brain. In addition, our data suggested that the cell-surface expression of SIRPα was not decreased in microglia from aged mice. Thus, the age-dependent dysfunction of CD47–SIRPα signals was unlikely. The increase in the number of CD11c$^+$ microglia in aged brains

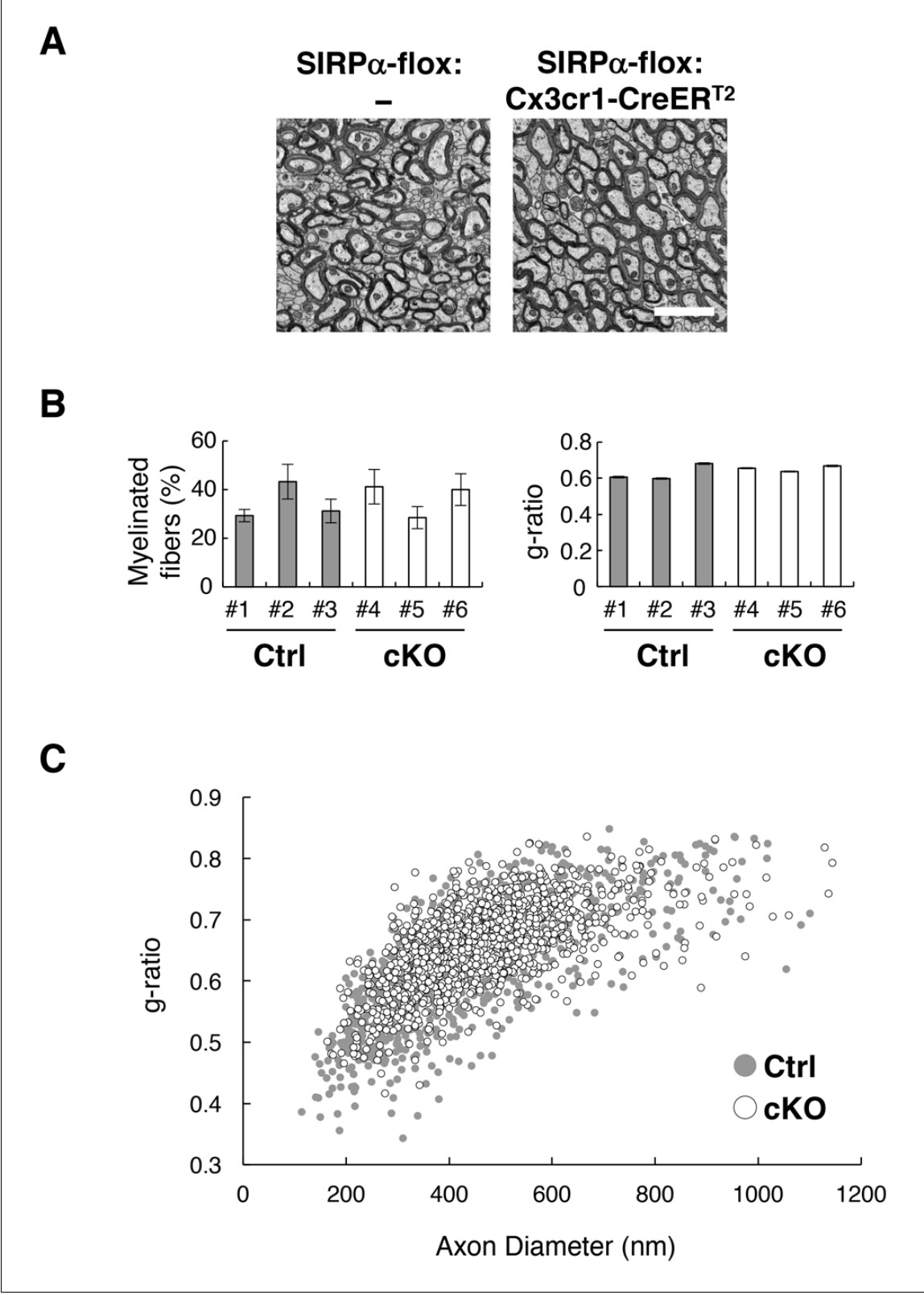

**Figure 7.** Normal myelin structure in microglia-specific SIRPα cKO mice. (**A**) Electron microscopic analysis of axon fibres in the anterior commissure of control (SIRPα-flox:—) and microglia-specific SIRPα cKO mice (SIRPα-flox:Cx3cr1-CreER[T2]) at 26–27 wks of age. Scale bar: 2 µm. (**B**) Quantitative data for the myelination ratio (percentage of myelinated axons in the total number of axons) and the g-ratio (a ratio of the inner axonal diameter to the total outer diameter of myelinated axons) in the anterior commissure of three control (#1–#3, *filled bars*) and three SIRPα cKO mice (#4–#6, *open bars*). The
*Figure 7 continued on next page*

*Figure 7 continued*

myelination ratio and the g-ratio were calculated from 9 to 10 images (109–465 axons/image) (n = 9–10) and 317–562 myelinated axons in 9–10 images (n = 317–562), respectively, for each mouse. Data are the means ± SEM. (C) Scatter plot showing total g-ratio data as a function of the axon diameters. Filled and open circles indicate control (n = 1147 axons from mice #1–#3) and SIRPα cKO mice (n = 1437 axons from mice #4–#6), respectively.

DOI: https://doi.org/10.7554/eLife.42025.015

may be explained by the age-dependent accumulation of tissue damage that stimulates microglia activation beyond the suppressive ability of SIRPα.

We demonstrated that CD11c+ microglia were specifically increased in the white matter of mutant mice. Thus, the white matter may contain an endogenous factor that promotes the induction of CD11c+ microglia in mutant mice. One potential candidate is myelin. As reported, CD11c+-microglia were increased in control SIRPα-flox:— mice after Cpz treatment (*Poliani et al., 2015*). Thus, myelin damage effectively induces CD11c+ microglia even in the control mice. This suggests that myelin degradation products that are formed during homeostatic turnover of the myelin structure might stimulate the induction of CD11c+ microglia in CD47–SIRPα signal-deficient mice, whereas the effect of such homeostatic degradation of myelin may be suppressed by SIRPα in the normal mouse brain. Enhanced phagocytosis of myelin components by microglia may be involved in the induction of CD11c+ microglia in our mutant mice, because SIRPα negatively regulated phagocytosis in macrophages (*Matozaki et al., 2009*). CD11c+ microglia in SIRPα-deficient mice expressed CD68, CD14, and Dectin-1, which are markers of phagocytic cells (*Devitt et al., 1998*; *Fu et al., 2014*; *Shah et al., 2008*), supporting the notion that phagocytic activity was increased in these cells.

Several studies have suggested the importance of myelin phagocytosis by microglia for the induction of CD11c+ microglia during demyelination or aging. The myelin-damage-dependent induction of CD11c+ microglia was markedly suppressed in mutant mice lacking Trem2, a phagocytic receptor (*Poliani et al., 2015*). The emergence of CD11c+ microglia during demyelination was also suppressed in Cx3cr1 KO mice, in which the phagocytosis of myelin debris by microglia was severely impeded (*Lampron et al., 2015*). In the brain of aged mice, the accumulation of myelin debris and the phagocytosis of such components by microglia were increased in the white matter (*Safaiyan et al., 2016*), where CD11c+ microglia were characteristically observed in aged mice (*Hart et al., 2012*). Of note, the aging-induced expansion of microglia in the corpus callosum was suppressed in Trem2 KO mice (*Poliani et al., 2015*). Furthermore, an in vitro study suggested that the presence of SIRPα on microglia inhibited the phagocytosis of myelin membrane fractions through interactions with CD47 on myelin (*Gitik et al., 2011*). Thus, our present data support the model in which the lack of CD47–SIRPα signal upregulates the phagocytosis of microglia, which become hypersensitive to the homeostatic destruction of myelin structures during the normal turnover process, and thereby triggers the induction of CD11c+ microglia without damage.

Transcriptome analyses showed that the expression of TNF-α (*Tnfa*), a proinflammatory cytokine, was increased 2–3-fold in the brain monocytes isolated from CD47 KO mice. In addition, pathway analysis showed that other genes related to the TNF and NF-κ B signaling pathway, including RelA (*Rela*) and IKKB (*Ikbkb*), were specifically increased in the mutant brain cells. Thus, it is likely that the proinflammatory TNF axis is strengthened in microglia by the lack of CD47–SIRPα signal. In contrast, the Wnt signal pathway, which contributes to the proinflammatory transformation of microglia (*Halleskog et al., 2011*), was attenuated in the CD47 KO brain cells. The transcriptome analysis also showed that the expression of TGF-β1 (*Tgfb1*), an anti-inflammatory cytokine, and of brain-derived neurotrophic factor (*Bdnf*), a neuroprotective factor, was increased in the CD47 KO brain cells. Thus, it is not clear whether CD11c+ microglia in the mutant mice have pro- or anti-inflammatory roles.

The expression of genes that are characteristically expressed during the recovery phase from demyelination (*Holtman et al., 2015*; *Olah et al., 2012*; *Poliani et al., 2015*), such as *Itgax*, *Igf1*, *Lpl*, *Apoc1*, *Ch25h*, *Mmp12*, *Spp1*, and *Cst7*, was increased in both the white matter and the brain cells of CD47 KO mice, suggesting their upregulation in the white matter CD11c+ microglia, and also suggesting the potential of the CD11c+ microglia to promote white-matter tissue repair. Consistently, after 5 wks of Cpz feeding, both demyelination and microgliosis were significantly reduced in SIRPα cKO mice. Although the direct involvement of CD11c+ microglia in the reduction of

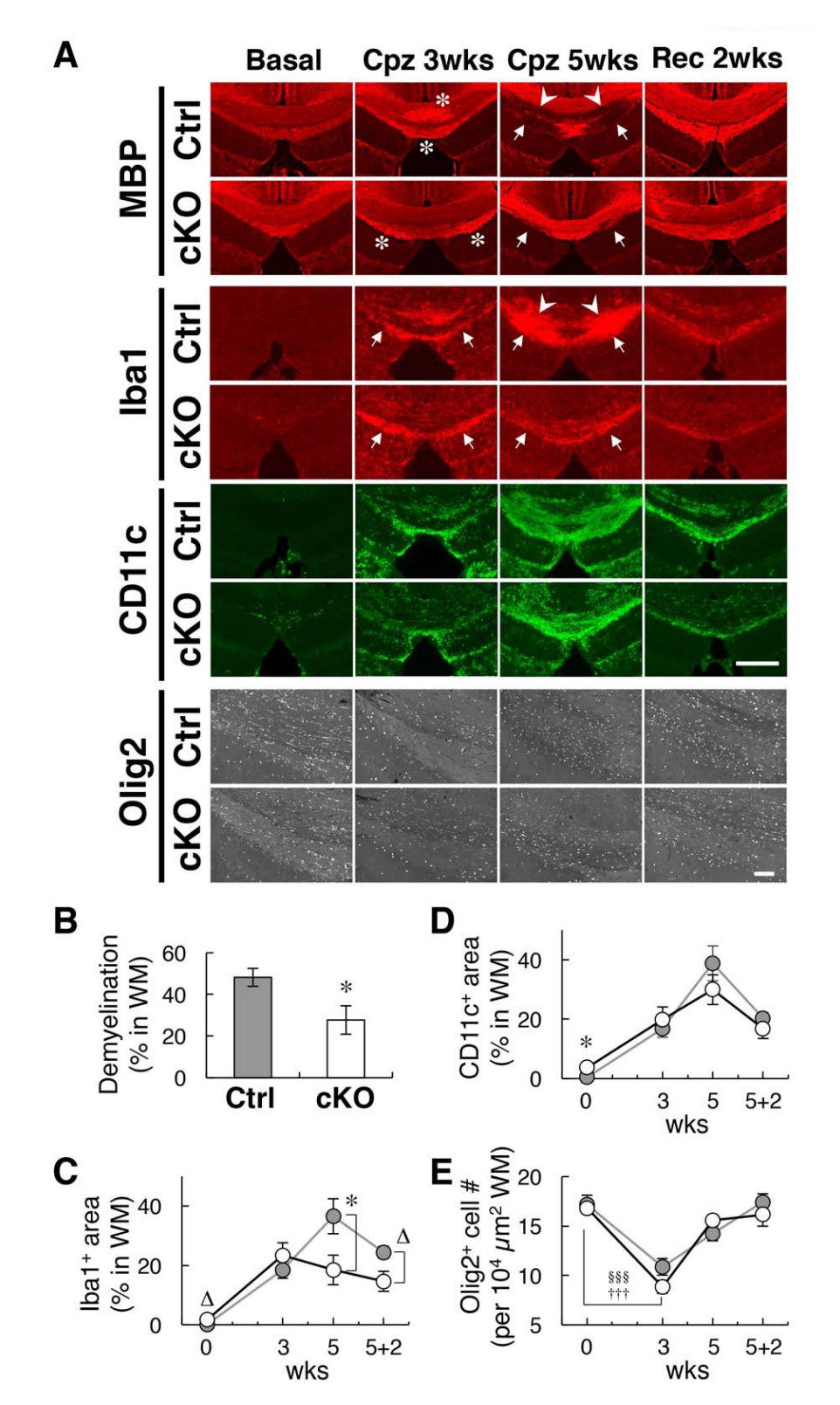

**Figure 8.** Alleviation of cuprizone-induced demyelination in SIRPα cKO mice. (A) Control (SIRPα-flox:— (Ctrl)) or SIRPα cKO (SIRPα-flox:Cx3cr1-CreER$^{T2}$ (cKO)) mice at 19–30 wks of age were fed a 0.2% (*w/w*) cuprizone (Cpz) diet. After 3 or 5 wks of Cpz feeding (Cpz 3 wks, Cpz 5 wks), brain samples were prepared. Other groups of mice were returned to a normal diet after 5 wks of Cpz treatment and allowed to recover for 2 wks prior to the tissue analysis (Rec 2 wks). Mice fed a normal diet without Cpz for 3 or 7 wks were analysed as controls (Basal). Brain sections were subjected to
*Figure 8 continued on next page*

*Figure 8 continued*
immunofluorescence staining with specific antibodies for MBP, Iba1 (*red*), CD11c (*green*), and Olig2 (*white*). Arrowheads and arrows indicate typical demyelination (MBP) or microgliosis (dense immunoreactivity for Iba1) in the corpus callosum (*arrowheads*) and hippocampal alveus (*arrows*). Asterisks indicate abnormally strong immunoreactivity of MBP at Cpz 3 wks. Scale bars: 500 µm for MBP, Iba1, and CD11c; 100 µm for Olig2. (B, C, D) Ratio of demyelination area to the white matter (WM) area (including corpus callosum and hippocampal alveus) in the analysed image at Cpz 5 wks (B), or the ratio of the Iba1-positive (C) or CD11c-positive (D) area to the WM area at Basal, Cpz 3–5 wks, and Rec 2 wks, were quantified by the immunoreactivity of MBP, Iba1, and CD11c, respectively. Data are the means ± SEM. A total of eight images obtained from four mice (0 — Basal; 3 — Cpz 3 wks; 5 — Cpz 5 wks), or a total of four images obtained from two mice (5 + 2 — Rec 2 wks) were analysed at each time point for each genotype. *p<0.05, Δp=0.06 (Welch's t-test). (E) The number of Olig2$^+$ oligodendrocytes were quantified in the WM area in the analysed image. Data are the means ± SEM. A total of 12–16 images obtained from four mice (0 — Basal; 3 — Cpz 3 wks; 5 — Cpz 5 wks), or a total of eight images obtained from two mice (5 + 2 — Rec 2 wks), were analysed for each genotype. §§§p<0.005 for Ctrl mice, †††p<0.005 for cKO mice (Welch's t-test) versus the basal value for the respective genotypes. No statistical difference was found between the two genotypes. Filled and open bars or circles in panels (B, C, D, E) indicate control and SIRPα cKO mice, respectively.
DOI: https://doi.org/10.7554/eLife.42025.016

demyelination damage in the mutant mice is not yet clear, CD11c$^+$ microglia are likely to be effective in protecting against Cpz-induced demyelination. IGF-1 and Spp1 were increased in the white matter and brain mononuclear cells of CD47 KO mice. These factors are derived from microglia and support the proliferation and differentiation of oligodendrocytes and myelination (*Olah et al., 2012*; *Selvaraju et al., 2004*; *Zeger et al., 2007*). CD11c$^+$ microglia in neonatal brains have been reported to be the major source of IGF-1 and to support primary myelination (*Wlodarczyk et al., 2017*). In our mutant mice, CD11c$^+$ microglia may facilitate white-matter repair through the function of these factors. Although Cpz feeding similarly induced CD11c$^+$ microglia in both control and SIRPα cKO mice, the preceding existence of CD11c$^+$ microglia in SIRPα cKO mice in the basal condition may enable these cells to respond to the damage earlier and may accelerate tissue repair through their protective effect. It is also possible that the protective function of CD11c$^+$ microglia may be enhanced in SIRPα-deficient CD11c$^+$ microglia when compared with wild-type CD11c$^+$ microglia. Further studies to compare the characteristics of CD11c$^+$ microglia in Cpz-treated control and SIRPα cKO mice will provide an insight into the mechanisms underlying the microglial SIRPα-dependent control of myelin damage.

During the recovery phase (after two weeks of feeding with normal chow), the size of the Iba1-positive area tended to be smaller in SIRPα cKO mice than in control mice, although this difference was not statistically signficant. This may be due to the relatively small number of experiments studying this time point. Further detailed studies are required to clarify the function of microglial SIRPα in the recovery phase of demyelination.

The loss and recovery of Olig2$^+$ cells were comparable in control and SIRPα cKO mice during de- and re-myelination. Cell death and restorative proliferation of the oligodendrocyte linage thus seems to occur to a similar extent in the two groups. The alleviated demyelination in SIRPα cKO mice may be explained by the faster differentiation and/or maturation of oligodendrocytes in these mice when compared with control mice, resulting in accelerated remyelination in the cKO mice. Another possibility is that both the cell death and restorative proliferation of oligodendrocytes were suppressed in SIRPα cKO mice, resulting in less damage than in control mice. In this case, the total number of Olig2$^+$ cells (the summation of surviving and proliferating Olig2$^+$ cells) in SIRPα cKO mice would be comparable to that in control mice, in which both the loss and restoration of Olig2$^+$ cells were much greater than in cKO mice, with greater demyelination.

Our data suggest a supportive role for CD11c$^+$ microglia in tissue repair. Consistently, the suppression of the emergence of CD11c$^+$ microglia by the genetic ablation of Trem2 was concurrent with exacerbated neuronal damage in demyelination model mice or neuronal death and amyloid deposition in AD model mice (*Poliani et al., 2015*; *Wang et al., 2015*). Since CD11c$^+$ microglia are characteristically observed in several neurodegenerative diseases (*Chiu et al., 2013*; *Holtman et al., 2015*; *Kamphuis et al., 2016*; *Remington et al., 2007*; *Wang et al., 2015*), the CD47–SIRPα signal may be widely involved in the pathology of degenerative brain diseases through the control of the protective function of microglia. The emergence of CD11c$^+$ microglia in aged brain might also be an adaptive reaction to the tissue damage caused by aging. It is also notable that the gene sets that were increased in the white matter of CD47 KO mice (i.e. *Itgax*, *Ipl*, *Cst7*, *Clec7a*, *Trem2*, and

*Tyrobp*) were similar to those increased in DAM (disease-associated microglia), a potential protective microglia subtype associated with neurodegenerative conditions such as AD (*Keren-Shaul et al., 2017*). CD47–SIRPα interactions may be a key mechanism for the induction of protective microglia such as DAM.

The reduction of demyelination in Cpz-treated SIRPα cKO mice is in contrast to the persistent demyelination that occurs after Cpz treatment in Trem2 KO mice (*Poliani et al., 2015*). In addition, the enhanced expression of *Itgax*, *Igf1*, *Lpl*, *Ch25h*, and *Apoc1* in CD47 KO mice contrasted markedly with that in Trem2 KO mice, in which the upregulation of these genes seen during demyelination in control mice was markedly suppressed (*Poliani et al., 2015*). The signaling mechanisms for SIRPα and Trem2 are contrasting. SIRPα activates tyrosine phosphatase Shp1/2, which binds to a phosphorylated ITIM in the cytoplasmic region of SIRPα, whereas Trem2 activates tyrosine kinase Syk, which binds to phosphorylated ITAM (immunoreceptor tyrosine-based activation motif) in the cytoplasmic part of DAP12, a membrane protein that forms a receptor complex with Trem2 (*Paradowska-Gorycka and Jurkowska, 2013*). SIRPα and Trem2 may reciprocally control microglia phagocytosis through the function of tyrosine phosphatase and kinase, respectively. Further analysis of the interaction between CD47–SIRPα signaling and Trem2–DAP12 signaling may be important in understanding microglial activation and phagocytosis and in identifying new therapeutic strategies to promote tissue repair in the damaged brain.

# Materials and methods

## Key resources table

| Reagent type (species) or resource | Designation | Source or reference | Identifiers | Additional information |
|---|---|---|---|---|
| Genetic reagent (*Mus musculus*) | *Sirpa*$^{-/-}$ (SIRPα KO) | PMID: 25818708 | RRID:MGI:5767142 | Dr. Takashi Matozaki (Kobe University, Japan) |
| Genetic reagent (*M. musculus*) | *Itgax-Venus* (CD11c–EYFP) | PMID: 15543150 | RRID:MGI:3835666 | The Jackson Laboratory (Bar Harbor, ME) |
| Genetic reagent (*M. musculus*) | *Cd47*$^{-/-}$ (CD47 KO) | PMID: 8864123 | RRID:MGI:1861955 | Dr. Pre-Arne Oldenborg (Umeå University, Sweden) |
| Genetic reagent (*M. musculus*) | *Sirpa*$^{flox/flox}$ | PMID: 25818708 | RRID:MGI:5767141 | Dr. Takashi Matozaki (Kobe University, Japan) |
| Genetic reagent (*M. musculus*) | *Cx3cr1*$^{tm2.1(cre/ERT2)Jung}$ (Cx3cr1–CreER$^{T2}$) | PMID: 23273845 | RRID:MGI:5467985 | Dr. Steffen Jung (Weizmann Institute of Science, Israel) |
| Genetic reagent (*M. musculus*) | *Itgax-cre* (CD11c–Cre) | PMID: 17591855 | RRID:MGI:3763248 | The Jackson Laboratory (Bar Harbor, ME) |
| Antibody | Rabbit polyclonal antibodies (pAbs) to Iba1 | Wako (Osaka, Japan) (Cat# 019–19741) | RRID:AB_839504 | IHC (1:250) |
| Antibody | Biotin-conjugated Armenian hamster monoclonal antibody (mAb) to mouse CD11c (clone HL3) | BD Pharmingen (Cat# 553800) | RRID:AB_395059 | IHC (1:200) |
| Antibody | Rat mAbs to CD68 (clone FA-11) | BioLegend (Cat# 137001) | RRID:AB_2044003 | IHC (1:50) |
| Antibody | PerCP-Cy5.5 conjugated rat mAb to CD45 (30-F11) | BD Biosciences (Cat# 550994) | RRID:AB_394003 | FCM (1:100) |
| Antibody | FITC-conjugated rat mAb to mouse CD11b (clone M1/70) | BD Biosciences (Cat# 553310) | RRID:AB_394774 | FCM (1:100) |
| Antibody | Rat mAb to mouse CD16/CD32 (clone 2.4G2) | BD Biosciences (Cat# 553141) | RRID:AB_394656 | FCM (1:400) |
| Antibody | Streptavidin-conjugated allophycocyanin (APC) | BD Biosciences (Cat# 554067) | RRID:AB_10050396 | FCM (1:200) |

*Continued on next page*

*Continued*

| Reagent type (species) or resource | Designation | Source or reference | Identifiers | Additional information |
|---|---|---|---|---|
| Antibody | PE conjugated rat mAb to mouse CD172a (SIRPa) (clone P84) | eBioscience (Cat# 12-1721-80) | RRID:AB_11149864 | FCM (1:100) |
| Antibody | PE conjugated rat mAbs to CD68 (clone FA-11) | BioLegend (Cat# 137013) | RRID:AB_10613469 | FCM (1:100) |
| Antibody | PE conjugated rat mAb to mouse CD14 (clone Sa14-2) | BioLegend (Cat# 123309) | RRID:AB_940582 | FCM (1:100) |
| Antibody | PE conjugated recombinant antibody (Ab) to Dectin-1 (REA154) | Miltenyi Biotec (Cat# 130-102-987) | RRID:AB_2651541 | FCM (1:5) |
| Antibody | PE conjugated recombinant Ab to CD47 (REA170) | Miltenyi Biotec (Cat# 130-103-108) | RRID:AB_2659745 | FCM (1:10) |
| Antibody | Rat mAb to myelin basic protein (MBP) (clone 12) | Merck (Cat# MAB386) | RRID:AB_94975 | IHC (1:500) |
| Antibody | Rabbit pAb to Olig2 | Immuno-Biological Laboratories (Gunma, Japan) (Cat# 18953) | RRID:AB_1630817 | IHC (1:400) |
| Antibody | Alexa Fluor 488 goat anti-rabbit IgG | Molecular Probes (Cat# A11034) | RRID:AB_2576217 | IHC (1:200) |
| Antibody | Cy3-conjugated AffiniPure Goat anti-rabbit IgG | Jackson Immuno Research (Cat# 111-165-144) | RRID:AB_2338006 | IHC (1:400) |
| Antibody | Cy3-conjugated AffiniPure Goat anti-rat IgG | Jackson Immuno Research (Cat# 112-165-167) | RRID:AB_2338251 | IHC (1:200) |
| Antibody | Cy3-conjugated AffiniPure Goat anti-mouse IgG | Jackson Immuno Research (Cat# 115-165-166) | RRID:AB_2338692 | IHC (1:400) |
| Antibody | Streptavidin, Alexa Fluor 488 conjugate | Molecular Probes (Cat# S11223) | RRID:AB_2336881 | IHC (1:400) |
| Commercial assay or kit | Black-Gold II myelin staining kit | Merck | Cat# AG105 | |
| Commercial assay or kit | Tyramide Signal Amplification (TSA) Biotin System kit | Perkin Elmer | Cat# NEL700A001KT | |
| Commercial assay or kit | RNeasy Mini kit | Qiagen | Cat# 74106 | |
| Commercial assay or kit | QuantiTect Reverse Transcription kit | Qiagen | Cat# 205313 | |
| Commercial assay or kit | QuantiTect SYBR Green PCR kit | Qiagen | Cat# 204143 or 24163 | |
| Commercial assay or kit | GeneChip 3'IVT Express Kit | Affymetrix | Cat# 901228 or 901229 | |
| Commercial assay or kit | Ovation Pico WTA system V2 | NuGEN | Cat# 3302–12/−60/–A01 | |
| Commercial assay or kit | Encore Biotin Module | NuGEN | Cat# 4200–12/−60/–A01 | |
| Chemical compound, drug | Tamoxifen | Toronto Research Chemicals Inc. | Cat# T006000 | |
| Chemical compound, drug | 4',6-Diamidino-2-phenylindole | Nacalai Tesque (Kyoto, Japan) | Cat# 11034–56 | |

*Continued*

| Reagent type (species) or resource | Designation | Source or reference | Identifiers | Additional information |
|---|---|---|---|---|
| Chemical compound, drug | Percoll | GE Healthcare Life Science (Uppsala, Sweden) | Cat# 17089102 | |
| Chemical compound, drug | Sepasol RNA I | Nacalai Tesque (Kyoto, Japan) | Cat# 09379–55 | |
| Chemical compound, drug | cuprizone | Sigma | Cat# C9012-25G | |
| Software, algorithm | FlowJo 8.8.4 software | Tree Star Inc. | RRID:SCR_008520 | |
| Software, algorithm | iTEM software | Olympus SIS | | |
| Software, algorithm | ImageJ | PMID: 22930833 | RRID:SCR_003070 | |
| Software, algorithm | Database for Annotation, Visualization and Integrated Discovery (DAVID) ver. 6.8 | PMID: 19033363 PMID: 19131956 | RRID:SCR_001881 | |

## Animals

*Sirpa*$^{-/-}$ (SIRPα KO), *Cd47*$^{-/-}$ (CD47 KO), *Sirpa*$^{flox/flox}$ (SIRPα-flox), and *Cx3cr1*$^{tm2.1(cre/ERT2)Jung}$ (Cx3cr1-CreER$^{T2}$) mice were generated as described previously (*Oldenborg et al., 2000*; *Washio et al., 2015*; *Yona et al., 2013*). *Itgax-cre* (CD11c-Cre) Tg mice (B6.Cg-Tg(*Itgax-cre*)1-1Reiz/ J, #008068) (*Caton et al., 2007*) and *Itgax-Venus* (CD11c-EYFP) Tg mice (B6.Cg-Tg(*Itgax-Venus*) 1Mnz/J, #008829) (*Lindquist et al., 2004*) were obtained from Jackson Laboratory (Bar Harbor, ME). SIRPα KO mice were crossed with CD11c–EYFP Tg mice to generate *Sirpa*$^{-/+}$:CD11c-EYFP Tg mice. These mice were crossed with heterozygous *Sirpa*$^{-/+}$ mice, and the resulting homozygous SIRPα WT and KO mice harbouring the CD11c–EYFP transgene were analysed. To obtain SIRPα cKO mice, homozygous SIRPα-flox mice harboring the CD11c–Cre transgene or Cx3cr1–CreER$^{T2}$ targeted gene were crossed with homozygous SIRPα-flox mice, and the resulting SIRPα cKO (SIRPα-flox:Cx3cr1-CreER$^{T2}$, SIRPα-flox:CD11c-Cre) mice were analysed. Littermates carrying homozygous SIRPα-flox alleles but lacking Cre recombinase were used as controls. To induce Cre-dependent recombination, Cx3cr1–CreER$^{T2}$ mice were treated with tamoxifen (TAM) (Toronto Research Chemicals Inc., Ontario, Canada) at 8 wks of age. TAM was dissolved in ethanol and then diluted seven times with corn oil (Wako, Osaka, Japan) to make a 10 mg/ml solution, and 200 µL (2 mg) of this solution was injected subcutaneously once every 48 hr for five consecutive days (a total of 3 times). TAM-treated mice were analysed more than 8 wks after the treatment. All mice were bred and maintained at the Biore-source Center of Gunma University Graduate School of Medicine under specific pathogen–free conditions. Mice were housed in an air-conditioned room with a 12-h-light, 12-h-dark cycle. All animal experiments were approved by the Animal Care and Experimentation Committee of Gunma University (approval no. 18-015).

## Primary antibodies and reagents

Rabbit polyclonal antibodies (pAbs) to Iba1 were obtained from Wako (Osaka, Japan). Biotin-conjugated Armenian hamster monoclonal antibody (mAb) to mouse CD11c (clone HL3), PerCP-Cy5.5 conjugated rat mAb to CD45 (30-F11), FITC-conjugated rat mAb to mouse CD11b (clone M1/70), rat mAb to mouse CD16/CD32 (clone 2.4G2), and streptavidin-conjugated allophycocyanin (APC) were from BD Pharmingen (San Diego, CA). PE conjugated rat mAb to mouse CD172a (SIRPα) (clone P84) was from eBioscience (San Diego, CA). PE conjugated and unconjugated rat mAbs to CD68 (clone FA-11) and PE conjugated rat mAb to mouse CD14 (clone Sa14-2) were from BioLegend (San Diego, CA). PE-conjugated recombinant antibody (Ab) to Dectin-1 (REA154) and PE-conjugated recombinant Ab to CD47 (REA170) were from Miltenyi Biotec (Bergisch Gladbach, Germany). 4',6-Diamidino-2-phenylindole (DAPI) was obtained from Nacalai Tesque (Kyoto, Japan). Rat mAb to myelin basic protein (MBP) (clone 12) was from Merck Millipore (Billerica, MA). Rabbit pAb to Olig2 (18953) was from Immuno-Biological Laboratories (Gunma, Japan).

## Histological analysis

For immunohistochemistry, mice were anesthetised by the inhalation of isoflurane (Pfizer, New York, NY), given an intraperitoneal injection of pentobarbital (Nembutal 100 mg/kg; Dynabot, Tokyo, Japan), and then perfused transcardially with fixation buffer (4% paraformaldehyde in 0.1 M phosphate buffer (pH 7.4)). Brain, spinal cord, or spleen was dissected and fixed in the fixation buffer overnight at 4°C, then transferred to 30% sucrose in 0.1 M phosphate buffer (pH 7.4) for cryoprotection, embedded in OCT compound (Sakura Fine Technical, Tokyo, Japan), and rapidly frozen in liquid nitrogen. Frozen sections with a thickness of 10 or 20 μm were prepared with a cryostat, mounted on glass slides, air dried, and washed with phosphate buffered saline (PBS). Otherwise, free-floating sections were washed with PBS. All sections were then incubated for 30 min at room temperature in Tyramide Signal Amplification (TSA) blocking solution (Perkin Elmer, Norwalk, CT) and stained overnight at 8°C with primary antibodies diluted in primary antibody dilution buffer (PBS supplemented with 2.5% BSA and 0.3% Triton X-100). Sections were then washed with PBS and exposed to the corresponding secondary antibodies conjugated with the fluorescent dyes Cy3 (Jackson Immuno Research Laboratories, West Grove, PA) or Alexa Fluor 488 (Invitrogen, Carlsbad, CA) in secondary antibody dilution buffer (PBS supplemented with 1% BSA and 0.3% Triton X-100). Nuclei were also stained with DAPI. The detection of CD11c with biotin-conjugated primary antibody (HL3) was achieved by using a TSA Biotin System kit (Perkin Elmer) and streptavidin-conjugated Alexa Fluor 488 (Invitrogen) according to the manufacturer's protocol. Free-floating sections were mounted on glass slides after staining. All sections were mounted with mounting media (0.1 M Tris-HCl buffer (pH9) containing 5% 1,4-diazabicylo (2.2.2)-octane (DABCO), 20% polyvinyl alcohol (PVA), 10% glycerol) and covered with a cover glass. Fluorescence images were acquired with a fluorescence microscope BZ-X710 (Keyence, Osaka, Japan) or DM RXA (Leica, Wetzlar, Germany) equipped with a cooled CCD camera (Cool SNAP HQ; Roper Scientific, Trenton, NJ). Acquired digital images were analysed using ImageJ software (*Schneider et al., 2012*).

For myelin staining, frozen brain sections were mounted on a glass slide first, and then stained with a Black-Gold II myelin staining kit (Merck Millipore) according to the manufacturer's protocol. Images were acquired with a light microscope DM IRBE (Leica) equipped with a cooled CCD camera (Penguin 600 CL; Pixera Corp., Santa Clara, CA).

## Preparation of microglia and flow cytometry

Mononuclear cells including microglia were isolated by the method described by *Sierra et al. (2007)* with a minor modification. Briefly, brain or spinal cord was dissected from genotype- and age-matched mice euthanised by cervical dislocation after deep anaesthesia. Male mice were used unless otherwise noted. Spinal cords isolated from 1 to 3 mice were mixed and treated together. Tissues were rinsed in ice-cold Hank's Balanced Salt Solution (HBSS; Gibco, Grand Island, NY) and then homogenised in HBSS containing collagenase type 2 (37.5 U/ml; Worthington) and DNase I (45 U/ml; Sigma). Homogenates were then incubated at 37°C for 30 min, gently triturated with a Pasteur pipette, and incubated at 37°C for 10 min. Undigested material was removed by filtration through a 70 μm cell strainer (BD Biosciences), and the remaining cells were centrifuged at 1100 × $g$ for 5 min at 18°C. Collected cells were then suspended in 8 ml of 1 × HBSS containing 70% Percoll (GE Healthcare Life Science, Uppsala, Sweden), and then 4–5 ml of each was placed into a 15 ml tube. Cell suspensions were then overlaid consecutively with 4 ml HBSS containing 37% Percoll and 1 ml PBS. The resulting gradient was centrifuged at 200 × $g$ for 40 min at 18°C, after which cells at the interface of the bottom two layers (70%/37%) were collected, washed twice with HBSS, and subjected to flow cytometric analysis.

For flow-cytometry analysis, cells were first incubated with a mAb specific for mouse CD16/CD32 to prevent nonspecific binding of labelled mAbs against FcγR and were then labelled with specific mAbs conjugated with PE, FITC, PerCP-Cy5.5, or biotin. For the staining of CD11c, a biotin-conjugated mAb against mouse CD11c was detected with streptavidin-conjugated APC. Labelled cells were analysed by flow cytometry using a BD FACS Canto II flow cytometer (BD Biosciences, San Jose, CA). Cells were first gated on their forward (FSC) and side (SSC) scatter properties to discriminate putative monocytes from other events, and then were gated on CD45 and CD11b. The CD11b$^+$/CD45$^{dim/lo}$ fraction obtained was analysed as microglia. All data were analysed with FlowJo 8.8.4 software (Tree Star Inc., Ashland, OR).

## Quantitative PCR analysis

Total RNA was extracted from the whole brain, spinal cord, or dissected optic nerve and optic tract with the use of Sepasol RNA I (Nacalai Tesque) and an RNeasy Mini kit (Qiagen, Hilden, Germany). First-strand cDNA was synthesised from total RNA with the use of a QuantiTect Reverse Transcription kit (Qiagen), and cDNA fragments of interest were amplified by real-time PCR in 96-well plates (Roche Diagnostics, Mannheim, Germany) with the use of a QuantiTect SYBR Green PCR kit (Qiagen) or a FastStart SYBR Green Master (Roche Diagnostics, Mannheim, Germany) and a LightCycler 480 or 96 Real-Time PCR System (Roche Diagnostics). The amplification results were analysed with the use of LightCycler software and were then normalised on the basis of the *Gapdh* mRNA level in each sample. Primer sequences (forward and reverse, respectively) were as follows: *Tnfa*, 5′-CCCTCACA-CACTCAGATCATCTTCT-3′ and 5′-GCTACGACGTGGGCTACAG-3′; *Il1b*, 5′-CAACCAACAAGTGA TATTCTCCATG-3′ and 5′-GATCCACACTCTCCAGCTGCA-3′; *Il6*, 5′-TAGTCCTTCCTACCCCAA TTTCC-3′ and 5′-TTGGTCCTTAGCCACTCCTTC-3′; *Il10*, 5′-AGGCGCTGTCATCGATTTCT-3′ and 5′-ATGGCCTTGTAGACACCTTGG-3′; *Tgfb*, 5′-ACCATGCCAACTTCTGTCTG-3′ and 5′-CGGGTTGTG TTGGTTGTAGA-3′; *Itgax*, 5′-AGCTGTGTGGACAGTGATGG-3′ and 5′-TGCATGTGAGTCAGGAGG TC-3′; *Igf1*, 5′-TACTTCAACAAGCCCACAGGC-3′ and 5′-ATAGAGCGGGCTGCTTTTGT-3′; *Trem2*, 5′-CTTCCTGAAGAAGCGGAATG−3′ and 5′-AGAGTGATGGTGACGGTTCC-3′; *Ccl3*, 5′-CAGC-CAGGTGTCATTTTCCT-3′ and 5′-CTGCCTCCAAGACTCTCAGG−3′; and *Gapdh*, 5′-TCCCACTC TTCCACCTTCGA-3′ and 5′-GTCCACCACCCTGTTGCTGTA-3′.

## Microarray analysis

Total RNAs were prepared from the white matter (optic nerve and optic tract) or brain mononuclear cells of WT or CD47 KO mice as described above. For the white matter, RNAs from five WT (13–15 wks of age) and four KO (12–15 wks of age) genotype-matched different male animals were pooled and subjected to the analysis. For the brain mononuclear cells, RNAs from seven WT (10–15 wks of age) and six KO (12–16 wks of age) male animals were pooled and analysed. Microarray analyses were performed by the Dragon Genomics Center of Takara Bio (Otsu, Japan). The quality of the RNA samples was confirmed by an Agilent 2100 Bioanalyzer (Agilent Technologies, Palo Alto, CA). Biotinylated complementary RNA (cRNA) was synthesized using the GeneChip 3IVT Express Kit (Affymetrix, Santa Clara, CA) from 250 ng of total RNA prepared from the white matter, or using the Ovation Pico WTA system V2 (NuGEN, San Carlos, CA) and the Encore Biotin Module (NuGEN) from 20 ng of total RNA prepared from the brain mononuclear cells. Following fragmentation, 10 μg of cRNA was hybridised for 16 hr at 45°C on the GeneChip Mouse Genome 430 2.0 Array (Affymetrix) with the use of a Hybridization, Wash and Stain Kit according to the GeneChip 3′IVT Express Kit User Manual (Affymetrix). GeneChips were then scanned by a GeneChip Scanner 3000 7G (Affymetrix) under the control of Affymetrix GeneChip Command Console Software (Affymetrix). Obtained data were processed using the Expression Console Software (Affymetrix).

## Transmission electron microscopy (TEM)

Sample preparation for TEM analysis was carried out as described previously (*Wilke et al., 2013*) with minor modifications. Mice were anesthetised as described above and were then perfused transcardially with 1.6% PFA and 3% glutaraldehyde in 0.1 M phosphate buffer (pH 7.4). Brain tissues were removed and fixed with 4% PFA in 0.1 M phosphate buffer overnight at 4°C. Brains were coronally sectioned into slices of 100 μm thickness, and these slices were incubated with a fixative containing 2% reduced $OsO_4$ and 1.5% potassium ferrocyanide in 0.1 M sodium cacodylate buffer (pH 7.4) for 1 hr on ice, with 1% thiocarbohydrazide solution for 20 min at room temperature and then with 2% OsO4 solution for 30 min. Sections were incubated in 1% uranyl acetate overnight at 4°C, incubated with Walton's lead aspartate solution for 75 min at 60°C, dehydrated by sequential treatment for 10 min in each of 50, 70, 80, 90, 95, and 100% ethanol, and then placed in a second solution of 100% ethanol. The sections were incubated with propylene oxide twice, with a 1:1 mixture of propylene oxide/Durcupan resin (Durcupan ACM, Fluka, Buchs, Switzerland) for 10 min and with Durcupan resin twice for 10 min each, and then cured on a slide glass at 60°C for 2 days. The cured resins containing the anterior commissure were trimmed out from the flat resins under a stereo microscope and re-embedded into resin blocks for ultrathin sectioning. Ultrathin sections were prepared at a 50 nm thickness using an ultramicrotome (Ultracut-T, Leica) and collected onto single slot

copper grids. The samples were examined at ×2500 magnification by a transmission electron microscope (H-7650, Hitachi, Tokyo, Japan) and digital electron micrographs were captured. The frequency of myelinated axons among all axons in the anterior commissure and the g-ratios of myelinated axons (the ratio of circumference of axon over myelin) were determined by the use of iTEM software (Olympus SIS, Münster, Germany).

## Cuprizone model of demyelination

Control (SIRPα-flox:—) or SIRPα cKO (SIRPα-flox:Cx3cr1CreER$^{T2}$) mice were fed a 0.2% (w/w) cuprizone (bis(cyclohexanone)oxaldihydrazone, Sigma) (Cpz) diet. Mice were sacrificed and their brains were processed for immunohistochemical analysis after 3 or 5 wks of Cpz treatment. Other groups of mice were returned to a normal diet after 5 wks of Cpz treatment and allowed to recover for 2 wks prior to immunohistochemical analysis. Mice fed a normal diet without Cpz for 7 wks were analysed as controls. Frozen brain coronal sections with a thickness of 20 μm were prepared and stained with specific antibodies for MBP (Myelin Basic Protein), Iba1, CD11c, and Olig2. Coronal slices from each mouse at approximate levels −1.9 to −2.0 mm from the bregma (*Paxinos and Franklin, 1997*) were analysed by fluorescence microscopy. Area size of the white matter, demyelination area (weak or no MBP-immunoreactive area), Iba1-positive area, and CD11c-positive area, as well as the cell number of Olig2$^+$ cells, were quantified by ImageJ software (*Schneider et al., 2012*). The white matter area, including corpus callosum and hippocampal alveus, was visually determined on the image. The ratios of the area size of demyelination area, the Iba1-positive area and the CD11c-positive area to that of the white matter region, as well as the cell density of Olig2$^+$ cells in the white matter area, were calculated from the quantified data and expressed as percentages. An example image for the quantification of demyelination size is shown in *Supplementary file 5*.

## Statistical analysis

Data are presented as the means ± SEM and were analysed using the Welch's t-test. A *P* value <0.05 was considered statistically significant.

## Data availability

The microarray data have been deposited to the Gene Expression Omnibus database (https://www.ncbi.nlm.nih.gov/geo/) (accession numbers: GSE118804 and GSE118805 for the white matter and the brain mononuclear cells, respectively).

# Acknowledgments

We thank E Urano and T Maegawa for technical assistance. This work was supported by a Grant-in-Aid for Scientific Research on Innovative Areas ('Brain Environment'), a Grant-in-Aid for Scientific Research (C), and a Grant-in-Aid for Challenging Exploratory Research from the Ministry of Education, Culture, Sports, Science and Technology of Japan, and by a grant from the Takeda Science Foundation of Japan.

# Additional information

### Funding

| Funder | Grant reference number | Author |
| --- | --- | --- |
| Ministry of Education, Culture, Sports, Science, and Technology | 26111703 | Hiroshi Ohnishi |
| Ministry of Education, Culture, Sports, Science, and Technology | 24111508 | Hiroshi Ohnishi |
| Japan Society for the Promotion of Science | 26670110 | Hiroshi Ohnishi |
| Japan Society for the Promotion of Science | 16K15189 | Miho Sato-Hashimoto |

| Japan Society for the Promotion of Science | 25430062 | Miho Sato-Hashimoto |
| Takeda Medical Research Foundation | | Miho Sato-Hashimoto |

The funders had no role in study design, data collection and interpretation, or the decision to submit the work for publication.

### Author contributions

Miho Sato-Hashimoto, Data curation, Funding acquisition, Investigation, Methodology, Writing—review and editing; Tomomi Nozu, Riho Toriba, Ayano Horikoshi, Miho Akaike, Kyoko Kawamoto, Ayaka Hirose, Yuriko Hayashi, Hiromi Nagai, Wakana Shimizu, Ayaka Saiki, Investigation, Writing—review and editing; Tatsuya Ishikawa, Ruwaida Elhanbly, Investigation, Methodology, Writing—review and editing; Takenori Kotani, Yoji Murata, Yasuyuki Saito, Per-Arne Oldenborg, Resources, Writing—review and editing; Masae Naruse, Koji Shibasaki, Resources, Methodology, Writing—review and editing; Steffen Jung, Conceptualization, Resources, Investigation, Writing—review and editing; Takashi Matozaki, Resources, Investigation, Writing—review and editing; Yugo Fukazawa, Methodology, Writing—review and editing; Hiroshi Ohnishi, Conceptualization, Data curation, Supervision, Funding acquisition, Validation, Investigation, Methodology, Writing—original draft, Project administration, Writing—review and editing

### Author ORCIDs

Miho Sato-Hashimoto (iD) http://orcid.org/0000-0002-0011-2753
Yoji Murata (iD) http://orcid.org/0000-0002-9576-7030
Yasuyuki Saito (iD) http://orcid.org/0000-0002-9291-1383
Koji Shibasaki (iD) http://orcid.org/0000-0003-2330-1749
Steffen Jung (iD) http://orcid.org/0000-0003-4290-5716
Takashi Matozaki (iD) http://orcid.org/0000-0002-4393-8416
Hiroshi Ohnishi (iD) http://orcid.org/0000-0002-2534-5449

### Ethics

Animal experimentation: All animal experiments were approved by the Animal Care and Experimentation Committee of Gunma University (approval no. 18-015).

### Decision letter and Author response

Decision letter https://doi.org/10.7554/eLife.42025.029
Author response https://doi.org/10.7554/eLife.42025.030

## Additional files

### Supplementary files

• Supplementary file 1. Gene expression changes in the white matter of CD47 KO mice. All genes changed more than 2-fold (|Log2 ratio| > 1) in the white matter of CD47 KO mice are listed. Probe set ID#, Affymetrix probe set ID number for Mouse 430 2.0 Genome Arrays; gene, gene description; symbol, gene symbol; accession, NCBI accession number; Log2 ratio, fold change expressed as Log2 (KO/WT).
DOI: https://doi.org/10.7554/eLife.42025.017

• Supplementary file 2. Gene expression changes in the brain mononuclear cells of CD47 KO mice. All genes changed more than 2-fold (|Log2 ratio| > 1) in the brain mononuclear cells of CD47 KO mice were listed. Probe set ID#, Affymetrix probe set ID number for Mouse 430 2.0 Genome Arrays; gene, gene description; symbol, gene symbol; accession, NCBI accession number; Log2 ratio, fold change expressed as Log2 (KO/WT).
DOI: https://doi.org/10.7554/eLife.42025.018

• Supplementary file 3. Immunostaining of SIRPα in the spleen of SIRPα cKO mice. Spleens were isolated from control (SIRPα-flox:—) and SIRPα cKO (SIRPα-flox:Cx3cr1-CreER$^{T2}$) mice 4 (*upper panels*) or 12 (*lower panels*) weeks after the administration of tamoxifen (TAM). Immunofluorescence staining with specific antibodies to SIRPα (*red*) are shown. The white pulp areas (WP) were surrounded by a white dotted line. Scale bar: 200 μm.
DOI: https://doi.org/10.7554/eLife.42025.019

• Supplementary file 4. Expression of CD47 on microglia prepared from SIRPα cKO mice. Cells were isolated from the spinal cord of control (SIRPα-flox:— (Ctrl)) or SIRPα cKO (SIRPα-flox:Cx3cr1-CreER$^{T2}$) mice at 25–28 wks of age, and the expression of SIRPα and CD47 on CD11b$^+$/CD45$^{dim/lo}$ microglia were analysed by flow cytometry. Expression profiles for SIRPα and CD47 in CD11b$^+$/CD45$^{dim/lo}$ microglia are shown. Filled and open traces indicate control and SIRPα cKO mice, respectively.
DOI: https://doi.org/10.7554/eLife.42025.020

• Supplementary file 5. An example of the quantification of demyelination size in the white matter of cuprizone -fed mice. A brain section prepared fromSIRPα-flox:— mice fed with a 0.2% (*w/w*) cuprizone diet for five weeks (Cpz 5 wks) were subjected to immunofluorescence staining with specific antibodies to MBP (*red* in the *left panel*). The white matter area analysed in the image is surrounded by a yellow line. Demyelination area with low MBP-immunoreactivity in the white matter area was shown in the right panel as filled (*black*) area. Ratio of area size (pixel number) of the demyelination area (*black* in the right panel) to that of the white matter area (*black* + *white* in the right panel) was calculated. alv, hippocampal alveus; cc, corpus callosum; cg, cingulum. Scale bar: 200 μm.
DOI: https://doi.org/10.7554/eLife.42025.021

• Transparent reporting form
DOI: https://doi.org/10.7554/eLife.42025.022

• Reporting standard 1. NC3Rs ARRIVE guidelines checklist.
DOI: https://doi.org/10.7554/eLife.42025.023

## Data availability

The microarray data discussed in this manuscript have been deposited to the Gene Expression Omnibus database (https://www.ncbi.nlm.nih.gov/geo/) (accession numbers: GSE118804 and GSE118805).

The following datasets were generated:

| Author(s) | Year | Dataset title | Dataset URL | Database and Identifier |
|---|---|---|---|---|
| Ohnishi H | 2018 | Expression data from mouse optic nerve and optic tract | https://www.ncbi.nlm.nih.gov/geo/query/acc.cgi?acc=GSE118804 | NCBI Gene Expression Omnibus, GSE118804 |
| Ohnishi H | 2018 | Expression data from mouse brain mononuclear cells | https://www.ncbi.nlm.nih.gov/geo/query/acc.cgi?acc=GSE118805 | NCBI Gene Expression Omnibus, GSE118805 |

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
