## [Decision Letter]

Thank you for submitting your article "Microglial SIRPα regulates the emergence of CD11c+ microglia and demyelination damage in white matter" for consideration by *eLife*. Your article has been reviewed by two peer reviewers, and the evaluation has been overseen by a Reviewing Editor and Huda Zoghbi as the Senior Editor. The following individuals involved in review of your submission have also agreed to reveal their identity: Peter A Calabresi (Reviewer #2); Anna Molofsky (Reviewer #3).

The reviewers have discussed the reviews with one another and the Reviewing Editor has drafted this decision to help you prepare a revised submission.

Summary:

This study examined the role of SIRP α (SIRPa) and CD47 signaling in both developmental myelination and remyelination in the mouse CNS, using SIRPa knockout, CD47 knockout and SIRPa microglial conditional knockout (using CX3CR1-CreER mice) to address the cell-type specific role of SIRPa in microglia. Global loss of either SIRPa or CD47, or conditional loss of SIRPa on microglia led to increased microglial numbers, activation, and expression of CD11c on white matter microglia. While baseline myelin properties and G ratio seemed normal in the conditional knockout, the authors report that cuprizone-induced demyelination was worse in the cKO animals. Together, these studies provide new insight into the mechanisms used to regulate the recruitment and activation of microglia following white matter injury.

Essential revisions:

1) The global and conditional knockouts phenocopy each other but the authors only show limited data for conditional knockouts in Figure 6 – staining for CD11c+/Iba1+ cells in the fimbria and normal developmental myelination. To more clearly demonstrate that the conditional knockout phenocopies the global knockout they should expand this data to show CD11c+ staining in other brain and spinal cord regions as demonstrated with the global knockout in Figure 1 and 2. It would be nice to see flow data on the CD11c+ population as for conditional knockouts, similar to what was shown in Figure 3 for SIRPa KO.

2) In Figure 6, g-ratios would be more clearly demonstrated with a typical g-ratio plot instead of histogram.

3) The authors should examine CD47 expression in wild-type compared to conditional knockouts to determine if CD47 is downregulated without SIRP interaction or if it is upregulated as compensatory response. If there is a reliable antibody, demonstrating this on different cell types such as neurons, microglia and oligodendrocytes, that would be ideal, but if not, flow data as shown in Figure 4C would be interesting for determining if there is a change in distribution of CD47+ microglia in absence of SIRPa.

4) For SIRPa conditional knockout with CD11c-Cre, would the authors elaborate on why they do not see similar phenotype as in CX3Cr1-Cre, which delete in monocytes, macrophages, mast cells and some cDCs? Do CD11c+ microglia down-regulate SIRPa and is that important prior to progression to a more activated state? Do these mice have a phenotype in the setting of demyelinating injury like was shown for the SIRPa CX3CR1 conditional knockouts?

5) It would be useful to see more data on the quantification of demyelination and gliosis in the cuprizone treated conditional knockouts in Figure 7. Some of the time points involved only two animals, which seems highly underpowered. It was not entirely clear what was being quantified from the images and graphs shown in Figure 7. The authors provide histograms to indicate demyelination and microgliosis but it is unclear according to the methods and figure legends how this data was quantified and what the axes represent. Some of the n values are low- 2 animals for certain genotypes and should be expanded. The MBP staining intensity is unclear with the gray scale images and should be more clearly illustrated with either fluorescent images or black-gold staining. Another method to quantify demyelination and recovery in the cuprizone model is to microdissect callosa and quantify MBP protein on a WB. Quantification of CD11c+ cells at all time-points should also be provided. Olig2 cells counts provide a measure of total oligodendrocytes but does not address whether by influencing the amount of activated microglia if there is any difference in the proportion of mature or immature oligodendrocytes which could be quantified with a mature marker such as CC1 and OPC markers such as PDGFRa. The text stating Olig2 is a marker of oligodendrocyte progenitor and lineage undergoing terminal differentiation into mature oligodendrocytes should be changed since Olig2 is more of a pan-oligodendrocyte marker.

6) The authors show that CD11c expression in white matter is altered in the SIRPa deficient animals, but so might any number of other markers which they have not examined. It would be helpful if this dataset were expanded to provide a broader overview of the immunological phenotype of these animals.

7) The statistical method used to analyze cell counts should be further clarified. Student's t-test would not be the appropriate statistical analysis as it assumes equal variance. Welch's t-test would be more appropriate.

---

## [Author Response]

Summary:This study examined the role of SIRP α (SIRPa) and CD47 signaling in both developmental myelination and remyelination in the mouse CNS, using SIRPa knockout, CD47 knockout and SIRPa microglial conditional knockout (using CX3CR1-CreER mice) to address the cell-type specific role of SIRPa in microglia. Global loss of either SIRPa or CD47, or conditional loss of SIRPa on microglia led to increased microglial numbers, activation, and expression of CD11c on white matter microglia. While baseline myelin properties and G ratio seemed normal in the conditional knockout, the authors report that cuprizone-induced demyelination was worse in the cKO animals. Together, these studies provide new insight into the mechanisms used to regulate the recruitment and activation of microglia following white matter injury.

The reviewers mentioned that cuprizone-induced demyelination was “worse” in our SIRPα cKO mice. However, there must be some kind of mistake in this sentence. Please understand that cuprizone-induced demyelination was “alleviated” in the cKO mice.

Essential revisions:1) The global and conditional knockouts phenocopy each other but the authors only show limited data for conditional knockouts in Figure 6 – staining for CD11c+/Iba1+ cells in the fimbria and normal developmental myelination. To more clearly demonstrate that the conditional knockout phenocopies the global knockout they should expand this data to show CD11c+ staining in other brain and spinal cord regions as demonstrated with the global knockout in Figure 1 and 2. It would be nice to see flow data on the CD11c+ population as for conditional knockouts, similar to what was shown in Figure 3 for SIRPa KO.

According to the reviewers’ comment, we showed CD11c^+^ staining of the brain and spinal cord sections prepared from the microglia-specific SIRPα conditional knockout (cKO) mice in the revised manuscript (new Figure 6B-E) as demonstrated with the SIRPα null knockout (KO) mice in the original Figure 1 and 2. We also performed new experiments to measure the Iba1^+^ and CD68^+^/Iba1^+^ cells in the fimbria (new Figure 6B and C) as demonstrated with the SIRPα null KO mice in the original Figure 1A and B. In addition, to respond to the reviewers’ comment, we added new flow data to show the increased expression of CD14, Dectin-1, and CD68 on the CD11c^+^ population in SIRPα cKO mice (new Figure 6—figure supplement 1) as shown for SIRPα null KO mice in the original Figure 3. We described these new results in the revised manuscript (subsection “Induction of CD11c^+^ microglia in the brain white matter of microglia-specific SIRPα-deficient mice”, second paragraph).

2) In Figure 6, g-ratios would be more clearly demonstrated with a typical g-ratio plot instead of histogram.

To respond to the reviewers’ comment, we rearranged a new scatter plot of g-ratio as function of axon diameters (new Figure 7C). The new scatter plot showed that the distribution profiles of g-ratio are comparable between two genotypes. We described this point in the revised manuscript (subsection "Alleviation of cuprizone-induced demyelination in the brain white matter of microglia-specific SIRPα-deficient mice”, first paragraph).

We would prefer to leave the original histogram (right panel in the original Figure 6E) in the revised manuscript (right panel in the new Figure 7B) as it was, since this graph provides the information regarding the data variation among three mice examined. However, if the reviewers strongly recommended to remove the histograms, we will change this.

3) The authors should examine CD47 expression in wild-type compared to conditional knockouts to determine if CD47 is downregulated without SIRP interaction or if it is upregulated as compensatory response. If there is a reliable antibody, demonstrating this on different cell types such as neurons, microglia and oligodendrocytes, that would be ideal, but if not, flow data as shown in Figure 4C would be interesting for determining if there is a change in distribution of CD47+ microglia in absence of SIRPa.

Ubiquitous expression of CD47 makes it difficult to examine the cell type-specific expression of CD47 by immunofluorescence staining with our antibodies. To respond to the reviewers’ comment, we examined the expression of CD47 in microglia prepared from SIRPα KO mice by flow cytometory as in the original Figure 4C for CD47 KO mice (new Figure 4D). As the reviewerspredicted, the expression of CD47 was increased in SIRPα-deficient microglia. These data suggest that lack of CD47-SIRPα signal may result in the upregulated expression of CD47 or stabilization of CD47-SIRPα complex in microglia. In addition, these data also suggest a possible microglia-microglia interaction through CD47-SIRPα signal, or cis-interaction between CD47 and SIRPα on the same microglia. Supporting these hypotheses, we also found that the expression of CD47 was increased in microglia prepared from miroglia-specific SIRPα cKO mice. However, this experiment could not be repeated, because the number of SIRPα cKO mice we maintained was limited. Thus, we prefer to show this data as a preriminary result in the new Supplementary file 4. We showed our additional data in new Figure 4D and new Supplementary file 4 and rewrote the Results and Discussion in the revised manuscript (Results subsection “Induction of CD11c^+^ microglia in the white matter of CD47-deficient mice”; Discussion, second paragraph).

4) For SIRPa conditional knockout with CD11c-Cre, would the authors elaborate on why they do not see similar phenotype as in CX3Cr1-Cre, which delete in monocytes, macrophages, mast cells and some cDCs?

As reviewer commented, the Cx3cr1-CreER^T2^ transgene has been reported to induce tamoxifen (TAM)-dependent rearrangement of a floxed gene not only in microglia, but also in peripheral myeloid cells (monocytes and dendritic cells) and tissue macrophages (Kupffer cells). However, the peripheral cells are short-lived and are replaced by their progeny without rearranged gene within a few weeks after administraiton of TAM (Goldmann et al., 2013). In contrast, the rearranged gene is remained stable in microglia more than 10 weeks after TAM administration (Goldmannet al., 2013), because of their long lifespan of around 15 months (Füger et al., 2017). As a result, microglia-specific gene targeting is achieved in Cx3cr1-CreER^T2^ mice several weeks after TAM treatment. Thus, in our experiments, SIRPα cKO mice were analyzed more than 8 weeks after TAM treatment as described in Materials and methods (subsection “Animals”). In contrast, CD11c-Cre mice have been reported to exhibit very low recombination of a floxed gene in microglia with less than 7% (Goldmannet al., 2013). That is consistent with the absence of CD11c expression in the majority of the resident microglia. Thus, genetic ablation of microglial SIRPα has been achieved with Cx3cr1-CreER^T2^, but not with CD11c-Cre. We clarified this point in the revised manuscript (subsection “Induction of CD11c^+^ microglia in the brain white matter of microglia-specific SIRPα-deficient mice”, first and last paragraphs).

Do CD11c+ microglia down-regulate SIRPa and is that important prior to progression to a more activated state?

A small subset of resident microglia was CD11c-positive but was hardly detected in wild-type mouse brain as shown in Figure 1 and 2. So it is difficult to produce convincing data regarding the expression change of SIRPα in CD11c^+^ microglia. However, in the CD11c^+^ cell-specific SIRPα cKO mice, in which SIRPα was genetically ablated in the resident CD11c^+^ microglia, we could not detect an increase in the number of CD11c^+^ microglia (new Figure 6—figure supplement 2), suggesting, at least, that down-regulation of SIRPα in the resident CD11c^+^ microglia is not prior to the expansion of these microglia. In contrast, conditional knockout of SIRPα in microglia, the majority of which is CD11c-negative, is sufficient to induce CD11c^+^ microglia, suggesting that loss of SIRPα in CD11c-negative microglia induces the expression of CD11c, an active microglia marker, in these cells.

Do these mice have a phenotype in the setting of demyelinating injury like was shown for the SIRPa CX3CR1 conditional knockouts?

As described above, Cx3cr1-CreER^T2^ mice achieve microglia-specific recombination of a floxed gene, while CD11c-Cre mice exhibit very low recombination in microglia. Thus, the brain phenotype of CDl1c-Cre: SIRPα cKO mice is likely to be different from that of Cx3cr1-CreER^T2^: SIRPα cKO mice. Reviewers may want to eliminate the possible contribution of SIRPα knockout in peripheral cells, such as monocytes, macrophages and dendritic cells, to the reduced demyelination phenotype in Cx3cr1-CreER^T2^: SIRPα cKO mice. However, as mentioned above, SIRPα cKO mice were analyzed more than 8 weeks after TAM treatment in our experiments (subsection “Animals”), and, at that time, the peripheral cells have already been replaced by their progeny without rearranged gene. To make this point clear, we added new data in the revised manuscript (new Supplementary file 3). These data showed that the immunoreactivity of SIRPα in the SIRPα cKO mouse spleen, where SIRPα^+^ cDC and macrophages exist, was once reduced after TAM treatment, but recovered after several weeks, probably because of the short life span and continuous turnover of these cells. In contrast, the rearranged gene has remained stable in microglia, because of their long lifespan (Goldmannet al. 2013). Thus, it is unlikely that genetic ablation of SIRPα in peripheral cells are involved in the phenotype of Cx3cr1-CreER^T2^: SIRPα cKO mice. As mentioned above, we clarified this point in the revised manuscript (subsection “Induction of CD11c^+^ microglia in the brain white matter of microglia-specific SIRPα-deficient mice”, first paragraph).

5) It would be useful to see more data on the quantification of demyelination and gliosis in the cuprizone treated conditional knockouts in Figure 7. Some of the time points involved only two animals, which seems highly underpowered. It was not entirely clear what was being quantified from the images and graphs shown in Figure 7. The authors provide histograms to indicate demyelination and microgliosis but it is unclear according to the methods and figure legends how this data was quantified and what the axes represent. Some of the n values are low- 2 animals for certain genotypes and should be expanded.

According to the reviewers’ comment, we added data in new Figure 8 in the revised manuscript (Figure 7 in the original manuscript). In the additonal experiments, we fed control and SIRPα cKO mice with cuprizone diet or normal diet for 3 weeks. The treatment protocol was same as described in our original manuscript, while the age of mice used were slightly different from the experiment shown in the original Figure 7. As a result, data from four control and four cKO mice were involved at “basal condition (without cuprizone)” and at “3 and 5 weeks cuprizone feeding” in the new Figure 8. Unfortunately, we could not expand the data for the recovery phase of the demyelination, in which mice were allowed to recover from the demyelination for 2 weeks with normal diet after 5 weeks of cuprizone treatment because of the limitated number of SIRPα cKO mice we have maintained. Instead, to improve the reliability of the data, we analyzed independent images from multiple sections. We clarified this point and stated the necessity of further detailed examination of the recovery phase in the Discussion in the revised manuscript (ninth paragraph). If the reviewers strongly recommended to increase the data in the recovery phase, we would like to consider it again, while it will take quite some time to prepare the sample for the analysis.

According to the reviewers’ comment, we tried to improve the quantification method in the revised manuscript and clarify what we quantified from the images. In the original manuscript, we quantified area size of demyelination and microgliosis in the analyzed image. In the revised manuscript, we quantified ratio (%) of area size of demyelination area, Iba1-positive area, and CD11c-positive area to that of the white matter region in the image data. We explained the details of the quantification in the Materials and methods(subsection “Cuprizone model of demyelination”) and rewrote the figure legends (new Figure 8). We also showed an example for the quantification of demyelination size in new Supplementary file 5.

The MBP staining intensity is unclear with the gray scale images and should be more clearly illustrated with either fluorescent images or black-gold staining. Another method to quantify demyelination and recovery in the cuprizone model is to microdissect callosa and quantify MBP protein on a WB.

According to the reviewers’ comment, we replaced the grey scale images of MBP staining, as well as that of Iba1 and CD11c staining, to color images in the revised manuscript (new Figure 8). Olig2 staining images have been left as grey scale, because we feel that the identfication of Olig2^+^ cells is easier in the grey scale images than in color images.

Quantification of CD11c+ cells at all time-points should also be provided.

According to the reviewers’ comment, we quantified CD11c staining and show the result in new Figure 8D. In the graph, the ratio (%) of size of CD11c-positive area to that of the white matter region in the image were calculated as described above (subsection “Cuprizone model of demyelination”).

Olig2 cells counts provide a measure of total oligodendrocytes but does not address whether by influencing the amount of activated microglia if there is any difference in the proportion of mature or immature oligodendrocytes which could be quantified with a mature marker such as CC1 and OPC markers such as PDGFRa. The text stating Olig2 is a marker of oligodendrocyte progenitor and lineage undergoing terminal differentiation into mature oligodendrocytes should be changed since Olig2 is more of a pan-oligodendrocyte marker.

We agree with this reviewers’ comment. In the original manuscript, we described that “Olig2 is a marker of oligodendrocyte progenitor and lineage undergoing terminal differentiation into mature oligodendrocytes”, because the expression of Olig2 is decreased in the mature oligodendrocytes. However, decreased expression is not equal to the lack of Olig2 in the mature oligodendrocytes. Indeed, Olig2 is expressed in mature oligodendrocytes. According to the reviewers’ comment, we changed our text and clarified that the Olig2 is a pan-oligodendrocyte marker (subsection “Alleviation of cuprizone-induced demyelination in the brain white matter of microglia-specific SIRPα-deficient mice”, last paragraph).

6) The authors show that CD11c expression in white matter is altered in the SIRPa deficient animals, but so might any number of other markers which they have not examined. It would be helpful if this dataset were expanded to provide a broader overview of the immunological phenotype of these animals.

In our research, we focused on CD11c because the expression of CD11c sharply reflects the activation of microglia in various neurodegenerative diseases, as well as during postnatal development and normal aging. However, as the reviewers mentioned, expression change of immune marker molecules other than CD11c occurred in microglia in SIRPα-deficient mice. In addition to CD11c, we have examined the expression of immune markers, such as CD68, CD14, and Dectin-1, in our mutant mice by flow cytometry (CD14, Dectin-1, CD68) and immunohistochemistry (CD68) (new Figure 1, Figure 3, Figure 6). We have also examined the expression of several cytokines in the brain and spinal cord (new Figure 3—figure supplement 2). Furthermore, transcriptome analysis of white matter and microglial fraction revealed changes in the expression profile of immune related pathways in CD47 KO mice (new Figure 5). Thus, we believe that these data provide, at least to some extent, a broader overview of the immunological phenotype of our mutant mice.

7) The statistical method used to analyze cell counts should be further clarified. Student's t-test would not be the appropriate statistical analysis as it assumes equal variance. Welch's t-test would be more appropriate.

According to the reviewers’ comment, we reanalyzed all data by the Welch’s t-test throughout the manuscript, and clarified the statistical method used in the Materials and methods (subsection “Statistical analysis”) and figure legends in the revised manuscript.